

# Multi-model approach to quantify groundwater level prediction uncertainty using an ensemble of global climate models and multiple abstraction scenarios

Syed M. Touhidul Mustafa [1,*], M. Moudud Hasan[1], Ajoy Kumar Saha[1], Rahena Parvin Rannu[1], Els Van Uytven[2], Patrick Willems [1,2] and Marijke Huysmans [1]

[1]Department of Hydrology and Hydraulic Engineering, Vrije Universiteit Brussel (VUB), Pleinlaan 2, 1050 Brussels, Belgium

[2]Department of Civil Engineering – Hydraulics Section, KU Leuven, Kasteelpark 40 box 2448, 3001 Leuven, Belgium

* Correspondence to: Syed Md Touhidul Mustafa (syed.mustafa@vub.be)

**Abstract**

Worldwide, groundwater resources are under a constant threat of overexploitation and pollution due to
anthropogenic and climatic pressures. For sustainable management and policy making a reliable prediction of
groundwater levels for different future scenarios is necessary. Uncertainties are present in these groundwater
level predictions and originate from greenhouse gas scenarios, climate models, conceptual hydro(geo)logical
models (CHMs) and groundwater abstraction scenarios. The aim of this study is to quantify the individual
uncertainty contributions using an ensemble of 2 greenhouse gas scenarios (representative concentration
pathway 4.5 and 8.5), 22 global climate models, 15 alternative CHMs and 5 groundwater abstraction scenarios.
This multi-model ensemble approach was applied to a drought prone study area in Bangladesh. Findings of this
study, firstly, point at the strong dependence of the groundwater levels on the CHMs considered. All
groundwater abstraction scenarios showed a significant decrease in groundwater levels. If the current
groundwater abstraction trend continues, the groundwater level is predicted to decline about 5 to 6 times faster
for the future period 2026-2047 compared to the baseline period (1985–2006). Even with a 30% lower
groundwater abstraction rate, the mean monthly groundwater level would decrease by up to 14 m in the
southwestern part of the study area. The groundwater abstraction in the northwestern part of Bangladesh has to
reduce by 60% of the current abstraction to ensure sustainable use of groundwater. Finally, the difference in
abstraction scenarios was identified as the dominant uncertainty source. CHM uncertainty contributed about 23%
of total uncertainty. The alternative CHM uncertainty contribution is higher than the recharge scenario
uncertainty contribution, including the greenhouse gas scenario and climate model uncertainty contributions. It is
recommended that future groundwater level prediction studies should use multi-model and multiple climate and
abstraction scenarios.



**Keywords**
Multi-model ensemble approach; Groundwater modelling; Conceptual models; Climate change; Abstraction
scenarios; Uncertainty.
**1. Introduction**
Groundwater is one of the major sources of high-quality fresh water across the world and one of the most
important but scarce natural resources in many arid and semi-arid regions. However, these resources are under a
constant threat of overexploitation and pollution all over the world due to anthropogenic and climatic pressure.
Globally, groundwater provides 45 – 70 % of irrigation water (Döll et al., 2012; Shamsudduha et al., 2011;
Taylor et al., 2013; Wada et al., 2014, 2013; Wisser et al., 2008) and the use of groundwater is continuously
increasing. Overexploitation of groundwater for irrigation is worldwide one of the main causes of groundwater
level depletion (Mustafa et al., 2017a; Rodell et al., 2009; Scanlon et al., 2012; Wada et al., 2014). Climate
change will probably also have an impact on the future availability of the groundwater resources (Brouyère et al.,
2004; Chen et al., 2004; Goderniaux et al., 2011, 2009; Scibek et al., 2007; Taylor et al., 2013; van Roosmalen et
al., 2009; Woldeamlak et al., 2007).
Food security of Bangladesh is highly dependent on sustainable use of groundwater for irrigation. However, in
the northwestern part of Bangladesh, these resources are under a constant threat of overexploitation due to
anthropogenic pressure. Mustafa et al. (2017a) report that overexploitation of groundwater for irrigation is the
main cause of groundwater level decline in the northwestern part of Bangladesh. In this context, the government
of Bangladesh has plans to use more surface water instead of groundwater. However, the amount of groundwater
that can be sustainably used for irrigation is still unknown. Also, the probable impact of shifting to more surface
water use instead of groundwater is also unknown. Hence, research is needed to quantify the amount of
groundwater that can be abstract sustainably for irrigated agriculture in the northwestern part of Bangladesh.
Accurate predictions of groundwater systems, as well as sustainable water management practices, are essential
for policy making. Transient numerical groundwater flow models are used to understand and forecast
groundwater flow systems under anthropogenic and climatic influences. They provide primary information for
decision-making and risk analysis. However, the reliability of groundwater model predictions is strongly
influenced by uncertainties resulting from the model parameters, input data, and the CHMs structure (Refsgaard
et al., 2006). Also, formulation of unknown future conditions, such as climatic change scenarios and
groundwater abstraction strategies, increases the uncertainty in groundwater model predictions.



It is important to assess the different sources of uncertainty to ensure accurate prediction and reliable decision
support in sustainable water resources management. The conventional treatment of uncertainty in groundwater
modelling focuses on parameter uncertainty. Uncertainties due to model structure and due to scenario change are
often neglected (Gaganis and Smith, 2006; Rojas et al., 2010). However, many researchers have recently
acknowledged that the uncertainty arising from the CHMs structure has a significant effect on model prediction
(Neuman, 2003; Refsgaard et al., 2006). The incomplete and biased representation of the processes and the
complex structure of a geological system often result in uncertainty in model prediction (Refsgaard et al., 2006;
Rojas et al., 2008). Højberg & Refsgaard (2005) presented a case of a multi-aquifer system in Denmark by
building three different CHMs using three alternative geological assumptions. They found that CHMs structure
uncertainty dominated over parameter uncertainty when the models were used for extrapolation. Many studies
have recently suggested that uncertainty derived from the definition of alternative CHMs is one of the major
sources of total uncertainty, and the parameter uncertainty does not cover the entire uncertainty range
(Bredehoeft, 2005; Neuman, 2003; Refsgaard et al., 2006; Rojas et al., 2008; Troldborg et al., 2007). Therefore,
neglecting the CHM uncertainty may result in unreliable prediction and underestimate the total predictive
uncertainty.
Studies using a single CHMs may fail to adequately sample the relevant space of plausible CHMs. Single model
techniques are unable to account for errors in model output resulting from the structural deficiencies of the
specific model. Rojas et al. (2010) noted that a CHM is assumed to be correct when the model is calibrated and
validated successfully following an appropriate method as described by Hassan (2004a, 2004b). However, a
well-calibrated model does not always accurately predict the behaviour of the dynamic system (Van Straten and
Keesman, 1991). Bredehoeft (2005) presented different examples where data collection and unforeseen elements
challenged well-established CHMs. Choosing a single model out of equally important alternative models may
contribute to either type I (reject true model) or type II (fail to reject false model) model errors (Li and Tsai,
2009; Neuman, 2003).
Although the concept of using alternative CHMs is increasing applied among surface water modellers, in
groundwater modelling the use of multi-model methods are limited. Recently, some studies have used multi-
model methods in groundwater modelling to quantify the CHM uncertainty (Li and Tsai, 2009; Rojas et al.,
2010). However, conceptual model uncertainty arising from the simplified representation of the hydro(geo)logic
processes, geological stratification and/or boundary conditions has received less attention (Refsgaard et al.,
2006; Rojas et al., 2010). Rojas et al. (2010), investigated uncertainty related to alternative CHM structures and





recharge scenarios in groundwater modelling. However, the uncertainty arising from other sources such as
General Circulation Models (GCMs), Regional Circulation Models (RCMs), downscaling methods and
abstraction scenarios in groundwater flow modelling still needs to be included in such approaches.
Climate change may significantly impact groundwater recharge. Recharge is one of the major input data in
groundwater levels simulation. The future groundwater recharge is unknown, so it should be estimated based on
different future climate scenarios. The GCMs project different climate scenarios based on the greenhouse gas
emission scenarios (GHSs). The Special Report on the Emission Scenario-SRES (Nakicenovic et al., 2000) has
reported different GHG emission scenarios. Besides, there are many GCMs to predict climate scenarios, and
different GCMs use a different representation of the climate system (Flato et al., 2013; Gosling et al., 2011;
Teklesadik et al., 2017). That means that different GCMs develop different climate projections for a single GHG
emission scenario. Therefore, uncertainties arise in climate projections from GCMs and GHG emission
scenarios. Another important source of uncertainties in climate projection is the internal variability of the climate
system, i.e., the natural variability of the weather (Deser et al., 2012). Future climate change uncertainty arises
from three main sources: external forcing, climate models response and internal variability (Hawkins and Sutton,
2009; Tebaldi and Knutti, 2007). Using an ensemble of climate scenarios has become common practice in
analysis of climate change impact in the field of hydrology. Uncertainty analysis of groundwater simulations
related to climate change has received relatively limited attention (Goderniaux et al., 2009; Taylor et al., 2013).
Holman et al. (2012) recommended that climate scenarios from multiple GCMs or RCMs should be used to
predict the impact of climate change on groundwater. Recently, several researchers have studied the impact of
climate change on the groundwater system incorporating uncertainty from the input of different GCMs or RCMs
scenarios and different greenhouse gas emission scenarios (Ali et al., 2012; Dams et al., 2012; Jackson et al.,
2011; Neukum and Azzam, 2012; Stoll et al., 2011; Sulis et al., 2012). The uncertainty analysis is, however,
usually limited to the climatic part. Very recently, Goderniaux et al. (2015) included uncertainty related to model
calibration in predicting groundwater flow along with uncertainty from the GCMs and RCMs and downscaling
methods. However, the uncertainty arising from other sources, such as the model conceptualization and
abstraction scenarios, is not evaluated.
Groundwater levels are often heavily influenced by the groundwater abstraction rate. For example, in the Indian
subcontinent, groundwater abstraction has increased from 10-20 km$^3$/year to approximately 260 km$^3$/year during
the last 50 years (Shamsudduha et al., 2011). In the northwestern part of Bangladesh, about 97% of the total
groundwater abstraction is used for irrigated agriculture (Mustafa et al., 2017a; Shahid, 2009). Shahid (2011)



found an increasing trend in irrigation application rate in Boro rice, the major irrigated crop in the area. Details
on current groundwater abstraction, trends in the abstraction and irrigated area can be found in Mustafa et al.
(2017a). This increasing trend is ascribed to climate change. In contrast, improvement in agricultural water use
efficiency can reduce the water use in irrigated agriculture. Therefore, multiple abstraction scenarios should be
used to predict a reliable uncertainty band.
Existing literature on future groundwater level prediction uncertainty quantification has focused on hydrological
model calibration and climate model uncertainty considering one single CHM and parameter uncertainty. As far
as the authors are aware, little research has been done so far to quantify future groundwater level prediction
uncertainty considering the uncertainty arising from the CHM structure, climate change and groundwater
abstraction scenarios. This is the first attempt to evaluate the combined effect of CHM structure, the climate
change and groundwater abstraction scenarios on future groundwater level prediction uncertainty.
The general objective of this study is to quantify groundwater level prediction uncertainty in climate change
impact studies using a multi model ensemble, i.e. an ensemble of representative concentration pathways, global
climate models, multiple alternative CHMs and abstraction scenarios to provide probabilistic and informative
predictions of groundwater levels. The specific objectives to achieve the general goal of this study are to: (i)
quantify the groundwater level prediction uncertainties arising from the definition of alternative CHMs; (ii)
analyse the effect of climate change on the groundwater levels using ensemble global climate models and
estimate the uncertainty linked to climate scenarios; (iii) analyse the effect of groundwater abstraction scenarios
on the future groundwater levels; (iv) quantify the amount of water that can be abstracted sustainably for
irrigated agriculture in the northwestern part of Bangladesh (v) evaluate the combined effect of CHMs structure,
the climate change and groundwater abstraction scenarios on future groundwater level prediction uncertainty;
and (vi) compare the uncertainty arising from the alternative CHMs, climate scenarios and abstraction scenarios.
**2. Methodology**
**2.1 Study area**
The study area is located in the northwestern part of Bangladesh (Figure 1a). The study area is a subtropical
region with two distinct seasons: the dry winter season (November to April) and the rainy monsoon season (May
to October). The average annual precipitation amount varies between 1400 and 1550 mm but is not uniformly
distributed over the year (Supplementary materials: Figure SM-2). Almost 83% of the total annual amount
occurs in the monsoon season. The average temperature varies between 25–35 °C for March to June, and 9–15



°C for November to February. Groundwater depth in the study area is continuously increasing (Supplementary
materials: Figure SM-3). The study area consists of six northwestern districts (Rajshahi, Naogaon, C'Nawabganj,
Joypurhat, Bogra and Nator) and cover about 7112 km². In comparison to other districts of Bangladesh, these
districts are more affected by drought (Shahid and Behrawan, 2008). The study area is situated between latitude
24°19´0´´ N to 25°12´0´´ N and longitude 88°6´36´´ E to 89°31´12´´ E. The surface elevation in the study area
varies from 11 m to 40 m (Supplementary materials: Figure SM-1). There is a mild gradient towards the
southeast corner and this corner is close to a large wet-land.
The aquifer in the study area is comprised of several layers such as clay, loamy clay, fine sand, medium sand,
coarse sand and gravel with a dominance of medium to coarse sand (Figure 1c). The thickness of each
stratigraphic unit moreover varies spatially. The top layer consists of clay, clayey loam and fine sand with an
average thickness of 18 m. It is underlain by a 20 m thick medium sand layer. Below the medium sand layer, a
35 m thick layer of coarse sand and coarse sand with gravel is present. The upper aquifer is unconfined or semi-
confined with a thickness ranging from 10 m to 40 m (Asad-uz-Zaman and Rushton, 2006; Faisal et al., 2005;
Jahani and Ahmed, 1997; Michael and Voss, 2009a; Rahman and Shahid, 2004). The area is dominated by
agriculture and almost 80 % is crop land. Irrigated agriculture plays an important role in the food production and
security of Bangladesh, home to over 150 million people. In the northwestern part of Bangladesh irrigated
agriculture is the major user of groundwater and accounts for 97 % of total groundwater abstraction (Shahid,
2009). Overexploitation of groundwater for irrigation, particularly during the dry season, causes groundwater-
level decline in areas where abstraction is high and surface geology inhibits direct recharge to the underlying
shallow aquifer (Mustafa et al., 2017a).





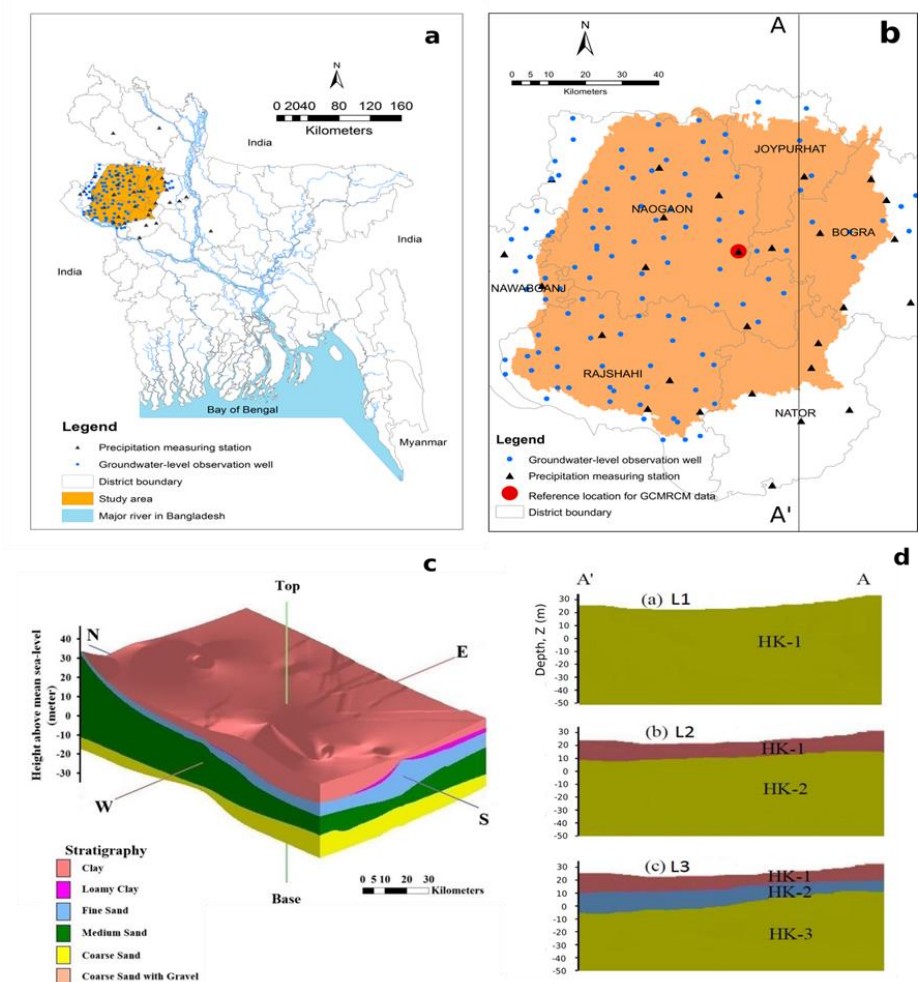


Figure 1: Description of the study area: (a) Location of the study area in the northwestern part of Bangladesh; (b)
study area with precipitation measurement stations (triangles) and groundwater observation wells (circles); (c)
stratigraphy of the study area; (d) cross-sectional (A-A') view of different models: (a) one-layered model (L1),
(b) two-layered model (L2), (c) three-layered model (L3).
**2.2 Data**
Thirty-two years (1979–2011) of weekly groundwater level and daily precipitation data of the Bangladesh Water
Development Board (BWDB) and Bangladesh Meteorological Department (BMD) were collected from the
Water Resources Planning Organization (WARPO), Bangladesh, for respectively 140 and 30 sites in the study
area. Available river discharge data of the BWDB for the existing small rivers within the study area were also





collected from WARPO. Daily minimum and maximum temperature, wind speed and other climatic data were
collected from the BMD for all the available stations in the country. Reference evapotranspiration ($ET_0$),
considered as potential evapotranspiration in this study, was calculated using the FAO Penman-Monteith
equation from the observed climatic data (Allen et al., 1998; Mustafa et al., 2017a).
The monthly observed groundwater head data of 50 observation wells were used for model calibration and
validation and are plotted in a box-plot (Supplementary materials: Figure SM-2). The groundwater levels vary
between 3 to 22 m above mean sea level (amsl) and display a clear seasonal variation. The groundwater level is
relatively low in April and high in October.
The hydraulic properties of the aquifers were selected based on observed data and previous reports on the
geology and lithology of the study area (Michael and Voss, 2009a, 2009b). Topography and borehole data were
collected from Barind Multipurpose Development Authority (BMDA), Bangladesh. The log data from twenty-
three boreholes within the study area were collected from BMDA.
The climate model data is available through the website of the Earth System Grid Federation
(https://esgf.llnl.gov).
**2.3 MODFLOW model**
Processing MODFLOW or PMWIN (Chiang and Kinzelbach, 1998) is a physically-based, fully-distributed, grid
based, integrated simulation system for modelling groundwater flow and pollution. PMWIN was designed as a
pre- and postprocessor for the groundwater flow model MODFLOW (Harbaugh and McDonald, 1996;
McDonald and Harbaugh, 1988) to bring various codes together in a simulation system. The MODFLOW model
is a physically-based, fully-distributed three-dimensional finite-difference numerical flow model developed by
the U.S. Geological Survey (USGS). MODFLOW solves the three-dimensional partial-differential groundwater
flow equation for porous media using a finite-difference method.
**2.4 Multi-step multi-model methodology**
A four-step methodology was used to achieve the objectives of the study (Figure 2). In the first step, the climate
model data for precipitation, minimum, mean and maximum temperature and $ET_0$ were extracted and
downscaled as explained in section 2.6. In the second step, monthly groundwater recharge was simulated using a
spatially distributed water balance model (WetSpass) (Abdollahi et al., 2017; Batelaan and De Smedt, 2001) for
the baseline period and for different scenarios as explained in sections 2.5.2 and 2.7. In the third step, 15
alternative conceptual hydrogeological models were constructed using different geological interpretations and



boundary conditions. All alternative CHMs were calibrated using observed groundwater level data. The
performance of each model was evaluated based on different performance evaluation coefficients and
information criterion statistics. The Bayesian model averaging (BMA) method was applied to obtain an average
prediction from the alternative models. Also, the performance of alternative models was evaluated based on the
maximum likelihood BMA weight of each model. The better performing models among the alternative models
were used to project groundwater levels under different climatic and abstraction scenarios. The averaged
projection and its uncertainty were estimated using BMA of the ensemble of alternative CHMs. In the final step,
climate change impact was assessed. The details of the different materials and methods of each step are
described in the following sections.





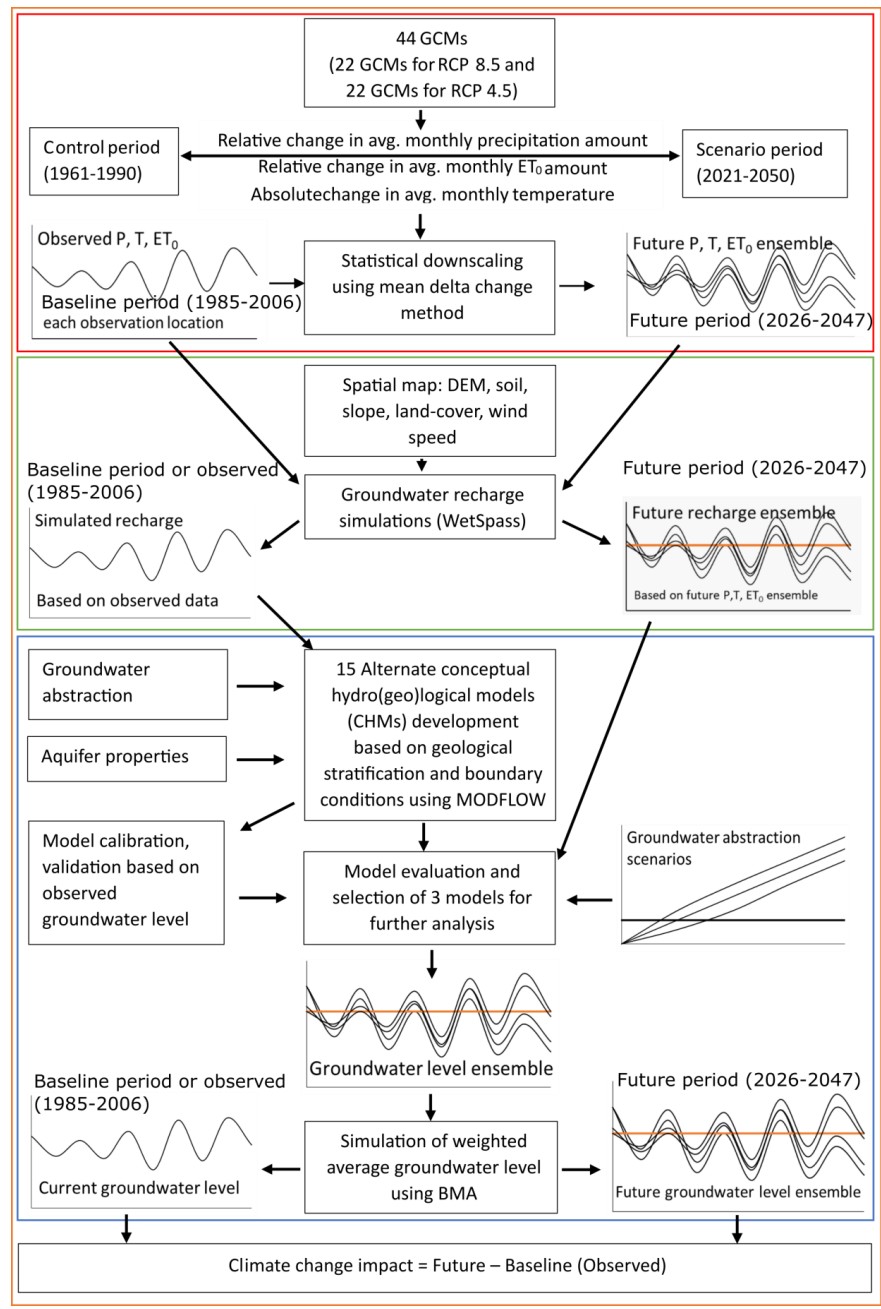


Figure 2: Multi-step multi-model methodology. GCM: General Circulation Model; RCP: Representative

Concentration Pathway; $ET_0$: potential evapotranspiration; P: precipitation; T: temperature; DEM: digital

elevation model; BMA: Bayesian model averaging.



### 2.5 Alternative conceptual groundwater flow models

To estimate the uncertainty due to the conceptualization of groundwater models, 15 different alternative CHMs were developed based on geological stratification and boundary conditions. The cross sectional (A-A') view of the models is shown in Figure 1d. First, three simplified alternative conceptual groundwater models were defined based on the geological stratification. The three models are a one-layered (L1), two-layered (L2) and three-layered (L3) model. In the one-layered model (L1), the entire model domain was considered as one hydro-stratigraphic unit and the hydraulic properties are assumed homogeneous and isotropic. The two-layered model (L2) consists of two layers where the average thickness of the top layer was 10 m (clay and loamy clay soil) and rest of the thickness was considered as the bottom layer. The model domain was divided into three different hydro-stratigraphic units to develop a three-layered model (L3). Below the top layer, a fine sand layer with an average thickness of 8 m was added in the three-layered model. The bottom layer of three-layered model consists of medium sand, coarse sand and coarse sand with gravel.

Boundary conditions strongly influence the CHM uncertainty (Wu and Zeng, 2013). They are often very uncertain, and, moreover, strongly influence the model results. Previous studies in the Bengal basin (Michael and Voss, 2009a, 2009b) identified a north to south groundwater flow direction. On the other hand, there is a large wetland at the southeastern corner of the study area as well as a large river (known as Ganges/Padma) within a few kilometers from the south boundary. Since exact boundary conditions were not known, based on above information, five different potential sets of boundary conditions were conceptualized and shown in Figure 3. For boundary condition B1, a no flow boundary condition was assumed on every side of the model. In other words, there is no interaction between the model domain and the environment (Michael and Voss, 2009a, 2009b). For boundary condition B2, a constant head boundary is assumed at the north side where most of the river branches originated assuming that groundwater flow direction is parallel to the river flow and perpendicular to the model boundary. No flow boundary conditions were assumed for all other sides. For boundary condition B3, a constant head boundary was considered on the north side like for B2 and southeastern side, i.e. the side where a large wetland is located. Boundary condition B4 is based on boundary condition B3. The constant head boundary in the southeastern part of the model was extended to the south part of the model domain in boundary condition B4 because the great Ganges/Padma river is very near to the south boundary. In boundary condition B5, a constant head boundary was considered at the north and northwestern boundary and also at the southeastern corner of the model domain based on the information that groundwater is flowing from north and northwestern to south (Michael and Voss, 2009a, 2009b). A constant head is assigned at the southeastern corner of the model domain





representing the Chalan Beel wetland as well. No-flow boundaries are assumed at the south and northeastern
boundaries since these boundaries are parallel to the groundwater flow direction (Michael and Voss, 2009a,
2009b). The long-term monthly average groundwater levels (normal) were considered as the constant
groundwater heads for the constant head boundary. As there is seasonal variability in the groundwater level of
this study area, every month was assigned a different constant groundwater head corresponding to the long-term
average groundwater level for that month.
In total, 15 alternative groundwater models were developed using 5 different boundary conditions and 3 different
layer types. A list of the 15 models is included as supplementary material (Table SM-1).

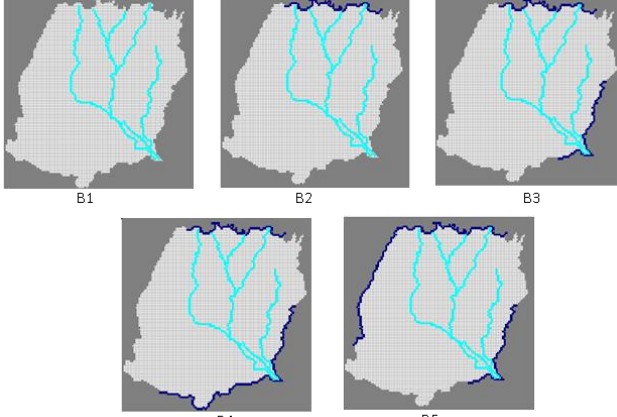


Figure 3: Boundary conditions used to develop alternative conceptual models (dark blue line indicates constant
head boundary). B1: no flow boundary; B2: constant head at north boundary; B3: constant head at north and
southeast boundary; B4: constant head at north, south and southeast boundary; B5: constant head at north,
northwestern and southeastern boundary.
**2.5.1 Model setup**
The BIock Centered Flow Package (BCF) of MODFLOW-96 within the PMWIN interface was used for
groundwater flow simulation. The study area covers an area of 7112 km$^2$ discretized into smaller cells having
rows and 118 columns. The grid cell dimension is 900 m x 900 m. All models are transient with a monthly
time step. A no-flow boundary is considered at the model domain bottom as the vertical groundwater flow is
restricted by the relatively impermeable hard rock below the aquifer in the study area. On the model top surface,
a spatially distributed recharge boundary is considered.





The initial groundwater heads correspond to a long-term average groundwater table obtained by running the
models in steady state conditions.
The range of hydrogeological parameter values was selected based on typical values for aquifer materials
(Domenico and Mifflin, 1965; Domenico and Schwartz, 1998; Johnson, 1967) and previous research findings in
the study area (Michael and Voss, 2009a, 2009b). They are listed in supplementary materials. Michael & Voss
(2009b) used $9.4 \times 10^{-5}$ m$^{-1}$ as specific storage value for Bengal basin. The initial specific storage was taken as
$9.4 \times 10^{-5}$ m$^{-1}$ when it is within the specific storage limits of the aquifer materials according to literature.
Otherwise, the initial specific storage was taken as the average of the maximum and minimum value of the
aquifer materials found in literature. The rivers in the study area are typically small and mainly driven by
precipitation runoff. Generally, there is no flow in the rivers during dry months (January to March). The "River
flow package" of MODFLOW was used to define rivers in the model domain and a third type boundary
condition was assumed for the rivers. Due to lacking field data for river bed materials, the river bed conductance
was obtained through manual calibration: river bed conductance is 0.18 m$^2$/s while riverbed thickness is 0.5 m.

**2.5.2 Simulation of spatially distributed groundwater recharge**

Spatially distributed monthly groundwater recharge was simulated using the WetSpass-M model (Abdollahi et
al., 2017; Batelaan and De Smedt, 2001) on the same grid as the groundwater flow (MODFLOW) model.
WetSpass-M is a physically based distributed model, in which the groundwater recharge is estimated from a
grid-based water balance. To allow land cover heterogeneity within each cell, every raster cell is split into four
fractions: vegetated, bare-soil, open-water and impervious. The water balances of each fraction are used to
calculate the total water balance of a raster cell, whereas recharge is calculated as the residual term of the water
balance for each cell. The inputs of the model are spatially distributed maps of land cover, soil texture,
topography, groundwater depth and climatic data. Precipitation (including of rainy days), ET$_0$, temperature and
wind speed were used as climatic information. Details on model setup and data preparation for groundwater
recharge calculation data can be found in Mustafa et al. (2017a). Monthly groundwater recharge was simulated
for twenty-two years (1985-2006) and considered as the baseline groundwater recharge.

**2.5.3 Groundwater abstraction estimation**

Groundwater abstraction for irrigation was calculated from the available data. Unfortunately, detailed
groundwater abstraction information e.g. amounts of water pumped from individual wells, co-ordinates of the
abstraction wells, capacity of the pumps or duration of pumping were not available. Hence, the groundwater




abstraction was assessed based on the irrigated area by shallow tube wells (STWs), deep tube wells (DTWs) and
other irrigation equipment. Upazila-wise (an upazila is the second lowest tier of regional administration in
Bangladesh) yearly seasonal groundwater abstraction for irrigation from the groundwater was calculated using
an empirical equation based on Boro rice irrigation requirements and the irrigated area. The irrigation water
withdrawal was considered as the total abstraction for each upazila. To obtain monthly abstraction for each
upazila, the calculated seasonal abstraction values are initially equally divided over the months of the dry
seasons (November to April). Also, as the location of the pumps is unknown, the total abstraction from each
upazila is initially considered uniformly distributed over the full upazila. Considering the individual upazila as
one zone of abstraction, a total of 34 abstraction zones were considered. Details on the irrigation data can be
found in Mustafa et al. (2017a) and Shamsudduha et al. (2015).
**2.5.4 Calibration and validation of alternative CHMs**
All alternative CHMs were calibrated for the period 1990-1994. Model parameters were estimated using manual
calibration and automatic calibration. During auto-calibration, PEST (Doherty, 1994) was used to optimize the
model parameter values.
The initial values, allowable ranges and optimized values of the parameters of the different models are given as
supplementary materials (Table SM-2, SM-3, SM-4). One-layered type models were calibrated for three
parameters: horizontal hydraulic conductivity, specific storage and specific yield. The two-layered and three-
layered models were calibrated for respectively 8 and 12 parameters. The process of selecting initial values and
the allowable range of the different parameters is described in section 2.5.1. The optimized specific storage of
the one-layered model with boundary condition-5 (L1B5) was $4.92 \times 10^{-05}$ m$^{-1}$. Michael & Voss (2009b) also
reported a similar specific storage value ($9.4 \times 10^{-05}$ m$^{-1}$) for the Bengal basin. However, different conceptual
models are suggesting different specific storage values within the typical values for aquifer materials depending
on the number of layers and boundary conditions (Table SM-2, SM-3, SM-4).
Using the optimized parameters, each of the alternative CHMs was validated for the period of 1995 to 1999.
**2.5.5 Model performance evaluation**
The performance of alternative conceptual groundwater models (CHMs) was evaluated using information
criterions, statistical indicators and by graphical presentation of simulated groundwater levels. Root Mean
Square Error (RMSE), Model Residual (error) Variance ($\sigma^2$), Nash-Sutcliffe Efficiency (NSE, Eq. 1) and Percent
Bias (PBIAS, Eq. 2) of the alternative CHMs were calculated using the formula reported by Moriasi et al.





(2007). Here, variance is defined as the mean squared error between observed and simulated value. The notation
of Mustafa et al. (2017b) has been followed.

$$\text{NSE} = 1 - \frac{\sum_{i=1}^{n}(O_i - S_i)^2}{\sum_{i=1}^{n}(O_i - \bar{O})^2} \qquad (1)$$

$$\text{PBIAS} = \left[\frac{\sum_{i=1}^{n}(O_i - S_i) * (100)}{\sum_{i=1}^{n} O_i}\right] \qquad (2)$$


Here, $O_i$ and $S_i$ are representing observed and simulated values respectively, $\bar{O}$ is the mean of $O_i$ and $n$ is the
number of observations.
NSE varies from $-\alpha$ to $+1$ and is dimensionless. NSE values closer to 1 mean better simulation efficiency. NSE
values $> 0.7$, $0.35 - 0.7$, $0.0 - 0.35$ and $< 0.0$ represent respectively, excellent, good, fair and poor performance.
The unit of PBIAS is percentage and values closer to zero mean better simulation capacity. Positive and negative
values are indicating respectively underestimation bias and overestimation bias (Gupta et al., 1999).
Information criteria are often used for model ranking (Zhou and Herath, 2017). Different information criteria
such as the Akaike Information Criterion (AIC), Corrected Akaike Information Criterion (AICc), Kashyap
Information Criterion (KIC) and Bayesian Information Criterion (BIC) were used to evaluate the alternative
CHMs.
The Akaike information criterion is defined as (Zhou and Herath, 2017):

$$AIC = n \ln(\sigma^2) + 2p \qquad (3)$$

$$AICc = n \ln(\sigma^2) + 2p + \frac{2p(p + 1)}{n - p - 1} \qquad (4)$$

$$\sigma^2 = \frac{SWSR}{n} \qquad (5)$$

Where n is the number of observations (same for all models), p is the number of model parameters = NPE+1,
NPE is the number of process model parameters and $\sigma^2$ is the residual variance. SWSR is the sum of weighted
squared residuals.
The Bayesian information criterion (BIC) and Kashyap information criterion (KIC) are defined in Eq. (6) and
(7), respectively (Zhou and Herath, 2017):

$$BIC = n \ln(\sigma^2) + p \ln(n) \qquad (6)$$

$$KIC = \left(n - (p - 1)\right)\ln(\sigma^2) - (p - 1)\ln(2\pi) + \ln|X^T \omega X| \qquad (7)$$

Where X is the sensitivity matrix (Jacobian matrix). The weighted factor $\omega$ applies when the errors are
independent from each other.
The different information criteria values were obtained from MODFLOW by running PEST in sensitivity
analysis mode. The best model among the alternative CHMs has a minimum information criteria value





(minimum AIC or AICc or BIC or KIC) (Zhou and Herath, 2017). A posterior model probability ($p_k$) was
calculated using Eq. (8) for each information criteria method for each alternative CHMs. The posterior model
probability was used to select the best CHMs. The better model corresponds to a larger posterior model
probability (Zhou and Herath, 2017).

$$p_k = \frac{e^{-0.5\Delta_k}}{\sum_{j=1}^{K} e^{-0.5\Delta_j}} \tag{8}$$

$$\Delta_k = AIC_k - AIC_{min} \tag{9}$$

Where $AIC_k$ is the AIC value for model k and $AIC_{min}$ is the minimum AIC values of all models. The value of $\Delta_k$
was also calculated for AICc, BIC and KIC.

**2.5.6 Bayesian model averaging**

Bayesian model averaging (BMA) was used to deduce more reliable predictions of groundwater levels than the
predictions produced by the individual groundwater models. Draper (1994) and Hoeting et al. (1999) present an
extensive overview of BMA. Recently, BMA has received attention of researchers of diverse fields because of
its more reliable and accurate predictions than other model averaging methods. Vrugt (2016) has developed a
model averaging MATLAB toolbox called MODELAVG for post-processing of forecast ensembles. The
MODELAVG has different model averaging methods including BMA and was used in this study. Details of the
model averaging method are described in the MODELAVG manual (Vrugt, 2016). The value of $\beta_{BMA}$
(maximum likelihood Bayesian weight) was used as a criterion to select the better performing models that have a
significant contribution in model averaging.
The general equation used to calculate the weighted average prediction in various model averaging strategies is
as follows:

$$\tilde{y}_j = \sum_{k=1}^{K} \beta_k D_{jk} \tag{10}$$

Where $D_{jk}$ is the bias corrected point forecasts of each model, k= {1,......, K} is model number and j= {1,.....n}
is the forecast number, $\tilde{y}_j$= { $\tilde{y}_1$,...., $\tilde{y}_n$} is the weighted average forecast for j[th] forecast number, β ={β₁,...., βₖ}
denotes the weight vector.

**2.6 Climate change scenarios**

The climate model data for precipitation, minimum, mean and maximum temperature are extracted for the grid
cells covering the reference location within the catchment. This reference location is set at 24.81° north and
88.95° east and is indicated by a red dot in Figure 1b. Using the FAO Penman-Monteith equation based on the
temperature from climate model data, $ET_0$ is calculated.





Within this case study, CMIP5 (Coupled Model Intercomparison Project Phase 5) climate model runs for RCP
4.5 and RCP 8.5 are considered (Taylor et al., 2012; Van Vuuren et al., 2011). RCP 8.5 is the highest RCP-based
greenhouse gas scenario (GHS) and considers a radiative forcing of 8.5 W/m² by 2100. The corresponding global
temperature rise ranges between 2.6 and 4.8°C. RCP 4.5 is a more intermediate scenario, whereby the radiative
forcing is limited to 4.5W/m² by 2100 and corresponding temperature rise between 1.4 and 3.1°C (IPCC, 2013).
The total climate model ensemble includes 44 runs, where the RCP 4.5 and RCP 8.5 sub-ensembles each include
22 runs. The considered climate model runs are listed in the supplementary materials (Table SM-7).
The goal number six of the United Nations (UN) sustainable development Goals (SDGs) states "Ensuring
availability and sustainable management of water and sanitation for all by 2030". Based on this information, the
climate change signals, are defined between 1975 and 2035, where the control and scenario period range
between 1961-1990 and 2021-2050, respectively. The precipitation and evapotranspiration changes are specified
on a relative basis, while for the temperature changes an absolute basis is considered. Using the delta change
method, the climate change signals are applied to the observed time series (Ntegeka et al., 2014). The delta
change method is a simple statistical downscaling method which applies mean monthly average changes (top
box of figure 2).

### 2.7 Future groundwater recharge scenario

The projected spatially distributed monthly groundwater recharge was simulated for the 44 projected time series
using the WetSpass-M model (Abdollahi et al., 2017; Batelaan and De Smedt, 2001) as explained in section
2.5.2 and in Mustafa et al. (2017a). The baseline groundwater recharge was calculated for a period of 22 years
(1985–2006). Future groundwater recharge was simulated for the same number of years (2026–2047). Simulated
groundwater recharges of the baseline period were compared to the simulated future groundwater recharge to
estimate the combined influence of the greenhouse gas scenarios or representative concentration pathways,
climate models and internal variability.

### 2.8 Development of future groundwater abstraction scenario

It is challenging to estimate future groundwater abstraction scenarios because it largely depends on human
activities as well as on climate. In this study, we have developed different future abstraction scenarios. The
groundwater abstraction data of the study area show a linearly increasing trend during 1985 to 2006 (Figure SM-
4: Supplementary materials). The increasing rate is different in different groundwater abstraction zones. The
average groundwater abstraction rate in 2006 was about five times higher than that in 1985. A similar increasing



trend in groundwater abstraction in the study area was also found by Mustafa et al., (2017a). Shahid (2011)
predicts an increasing trend in future irrigation application for Boro rice production due to climate change. He
also predicts that the length of Boro rice growing period may decrease in future which may lead to increased
cropping intensity in the area. Increased cropping intensity may increase the overall yearly groundwater
abstraction rate. Moreover, it is estimated that population of Bangladesh will increase from 145 million in 2008
to 182 million by 2030 (Qureshi et al., 2014). Thus, water use for food production will increase tremendously.
As groundwater is the major source of water in the study area, groundwater withdrawal rate will be much higher.
However, there has not been an effective groundwater abstraction policy before 2017. Recently, the Integrated
Minor Irrigation Policy 2017 and the Groundwater Management Law 2018 for agriculture have been proposed to
ensure sustainable irrigation management. Both the Integrated Minor Irrigation Policy 2017 and the
Groundwater Management Law 2018 have recommended to minimize the groundwater abstraction in the study
area to maintain sustainable groundwater abstraction. They also encourage to use surface water instead of
groundwater for the irrigation. Unfortunately, no quantitative or specific action for example how much
abstraction should be reduced, has been mentioned either in the proposed Integrated Minor Irrigation Policy
2017 or in the Groundwater Management Law 2018. The policy planning and management strategies should be
updated based on the quantitative or specific information.
Groundwater abstraction can be reduced by improving agricultural water use efficiency. The agricultural water
use efficiency is extremely low in Bangladesh. On average, crops use only 25–30% of applied irrigation water
and the rest is lost due to inefficient irrigation systems (Karim, 1997; Mondal, 2010, 2005). Using efficient
irrigation distribution and application techniques can increase agricultural water use efficiency. The BMDA has
introduced a buried PVC pipe water conveyance system in the study area to increase conveyance efficiency to
more than 90%, whereas the national average value is 40% (Rahman et al., 2011). Alternate Wetting and Drying
(AWD) rice irrigation technique can save 30 to 70% of water compared to conventional irrigation methods
(Rahman and Bulbul, 2015). Deficit irrigation in wheat cultivation in the study area can save 121–197 mm of
water per season (Mustafa et al., 2017b). Food habit changes and/or crop diversification may also have an impact
on crop water use efficiency.
Considering the uncertainties on the total groundwater abstraction amount, five different groundwater abstraction
scenarios are developed (**Error! Reference source not found.**Table 1). The first scenario is developed based on
the current increasing trend. The second scenario assumes an improved irrigation water use. As such the





conveyance efficiency will compensate the increasing future demand and the groundwater abstraction rate will
remain constant. In other words, this scenario considers the groundwater abstraction rate for 2010. The third,
fourth and fifth scenarios assume respectively 30%, 50% and 60% lower groundwater abstraction, where the
groundwater abstraction rate in 2010 was considered as a basis.
Table 1: Description of future groundwater abstraction scenarios.

| Groundwater abstraction scenario | Description |
|---|---|
| $P_{Linear}$ | Linear increase of groundwater abstraction rate based on current increasing trend |
| $P_{Constant}$ | Groundwater abstraction rate of 2010 assumed to be constant in future |
| $P_{Reduced\_30}$ | 30% less groundwater abstraction than in 2010 |
| $P_{Reduced\_50}$ | 50% less groundwater abstraction than in 2010 |
| $P_{Reduced\_60}$ | 60% less groundwater abstraction than in 2010 |


**2.9 Uncertainty estimation**
The spread of the 95% prediction interval was taken as the uncertainty band of the ensemble. The uncertainty
band was estimated using Eq. (11).

$$U_{band}^n = D_{97.5}^n - D_{2.5}^n \tag{11}$$

$$U_{avg} = \frac{1}{N} \sum_{n=1}^{N} U_{band}^n \tag{12}$$


Where $U_{band}^n$ is the uncertainty band of a time step, $U_{avg}$ is the average uncertainty band, N is the total number of
predictions, $D_{97.5}^n$ and $D_{2.5}^n$ represent the 97.5[th] and 2.5[th] percentile of the ensemble at a time step, respectively.
In the case of alternative CHM uncertainty quantification, the same abstraction and recharge scenarios of the
baseline period were used to simulate groundwater levels of the 22-year period. To quantify the recharge
scenario uncertainty, the groundwater level was simulated for 44 recharge scenarios by the best performing
groundwater flow model where the groundwater abstraction scenario was kept the same. The groundwater level
was simulated for 5 abstraction scenarios by the best performing groundwater flow model where the same





443 recharge scenario was used to estimate abstraction scenario uncertainty. The groundwater levels in 50

444 observation wells for a period of 22 years were used to estimate the spread of the 95% prediction interval.

445 The contribution of the different sources of uncertainty in future groundwater level prediction was calculated

446 considering all the probable combinations of the CHMs, recharge and abstraction scenarios. The average

447 prediction interval at each time step was calculated using the following equations:

$$U^n_{CM_{avg}} = \frac{1}{AS \times RS} \sum_{AS=1}^{AS} \sum_{RS=1}^{RS} U^n_{CM_{AS,RS}} \qquad (13)$$

$$U^n_{R_{avg}} = \frac{1}{K \times AS} \sum_{K=1}^{K} \sum_{AS=1}^{AS} U^n_{R_{K,AS}} \qquad (14)$$

$$U^n_{A_{avg}} = \frac{1}{K \times RS} \sum_{K=1}^{K} \sum_{RS=1}^{RS} U^n_{A_{K,RS}} \qquad (15)$$

448 Where, $U^n_{CM_{avg}}$, $U^n_{R_{avg}}$ and $U^n_{A_{avg}}$ represent the average prediction interval at each time step due to CHMs,

449 recharge scenario and abstraction scenario, respectively. The K, AS and RS represent the number of CHMs,

450 abstraction scenarios and recharge scenarios, respectively. The $U^n_{CM_{AS,RS}}$ is the prediction interval due to different

451 CHMs for a particular recharge and abstraction scenario. The $U^n_{R_{K,AS}}$ and $U^n_{A_{K,RS}}$ represent the prediction interval

452 due to different recharge scenario and abstraction scenario, respectively for a particular CHMs and

453 abstraction/recharge scenario.

454 **2.10 Data analysis**

455 For data analysis and plotting, different Matlab, R and Python packages were used such as Pandas (McKinney,

456 2010), Scipy, ggplot2, Numpy (Walt et al., 2011) and Matplotlib (Hunter, 2007). The null hypotheses for equal

457 distributions of simulated groundwater levels of alternative CHMs were tested using two-sample Kolmogorov-

458 Smirnov tests (Chakravarti and Laha, 1967). The nonparametric modified Mann-Kendal trend test (Hamed and

459 Rao, 1998) was conducted to detect trends in annual groundwater level and the slope was estimated using Sen's

460 method (Sen, 1968).

461 **3. Results and discussion**

462 **3.1 Groundwater levels simulation**

463 The simulated groundwater levels of each alternative groundwater flow model were compared to the observed

464 groundwater levels as well as to the simulated groundwater levels of the other models. The null hypotheses for



the equal distribution test between simulation results of alternative models in the calibration and validation
period were tested (Figure 4). A significant difference (significance level of 0.05 or p<0.05) between most of the
alternative model's simulation results was observed. This indicates that the use of different geological
stratifications and boundary conditions in groundwater flow models can result in significant differences in
groundwater levels prediction and confirms the finding of Rojas et al. (2010). In contrast, some of the models did
not predict statistically different results.

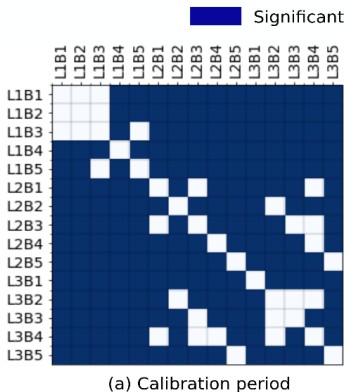
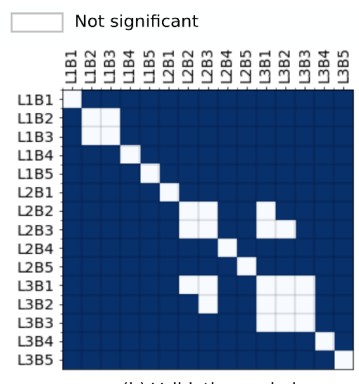

Figure 4: Significance of difference in simulation results for combinations of alternative conceptual models
(p<0.05, two sample K-S test) for (a) calibration and (b) validation period. L1, L2 and L3 are representing
respectively the one, two and three-layered model. B1, B2, B3, B4 and B5 are representing respectively
Boundary condition-1,2,3,4 and 5. For example: L1B1: One-layered model with Boundary condition-1, L3B5:
Three-layered model with Boundary condition-5.
**3.1.1 Goodness of fit of alternative CHMs**
Based on different statistical coefficients, the performance was different for alternative models, and the models
performed differently in the calibration and validation period (Supplementary materials: **Error! Reference**
**source not found.**Table SM-5).
Based on RMSE, $\sigma^2$ and NSE value, the L2B3 model was the best model in the calibration period, whereas in the
validation period it was L2B5. In general, the two-layered models had a relatively lower RMSE and $\sigma^2$ than the
one-layered and three-layered models.
In both the calibration and validation period, PBIAS was negative for one-layered models indicating that the
models were overestimating groundwater head. On the contrast, two-layered and three-layered models generally



underestimated the groundwater heads as PBIAS was positive in the calibration and validation period. The L2B5
and L2B4 model had the lowest bias in the calibration and validation period, respectively. Overall, the two-
layered models outperformed the one-layered and three-layered models in the calibration and validation period.
The simplified one-layered models have a comparatively higher bias in prediction. Comparatively, a large
number of processed parameters made the three-layered models over-parameterized. The three-layered models
performed better than the one-layered models during calibration, but they performed similarly in most of the
cases in the validation period. The performance of the two layered models also differed between calibration and
validation period. It is difficult to calibrate over-parameterized models efficiently (Willems, 2012), so the two-
layered models with eight calibrated parameters can be a balance between oversimplified and over-
parameterized models.
Figure 5 shows the scatter plot for model L2B5. One of the possible causes of the outliers in the scatter plot and
the differences in model performance between the calibration and validation period is the spatial and temporal
variation in groundwater abstraction. The zone-wise spatially distributed groundwater abstraction rate was one of
the most important input data in this study. In reality, groundwater abstraction varies spatially within those
zones. Agricultural and industrial areas abstract more groundwater than wetlands or forest areas. Moreover,
groundwater abstraction rate also depends temporally on cropping season and precipitation pattern. However, an
average constant groundwater abstraction rate was assumed for six months (from November to April) in the
model. For observation wells close to groundwater abstraction wells, drawdown by groundwater abstraction,
could affect the observed groundwater heads. This spatial and temporal difference in actual groundwater
abstraction and modeled groundwater abstraction caused spatial and temporal variation in simulated and
observed groundwater levels. The simplified representation of hydrogeological properties could be also a
possible cause of the difference between simulated and observed groundwater levels. For simplification, the
aquifer was assumed homogeneous but in reality, the aquifer is heterogeneous and this may affect groundwater
flow in the aquifer. Also measurement errors in observation data influence model performance.



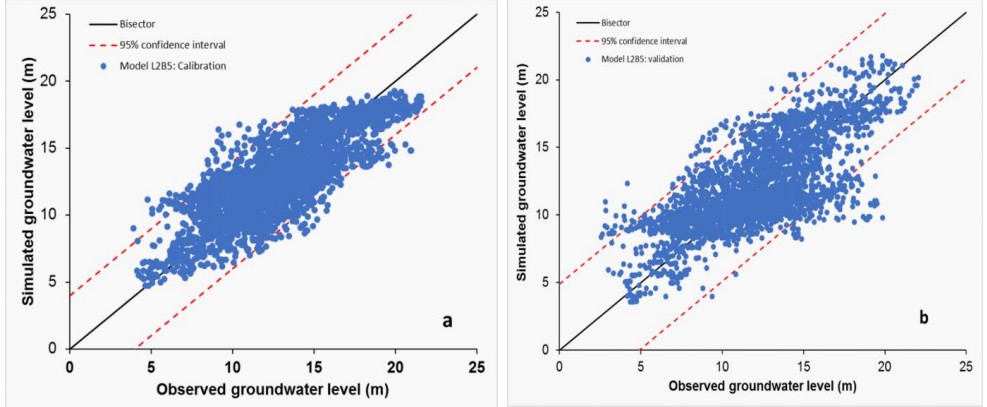


Figure 5: Scatter plot for the simulated versus observed groundwater level for Model L2B5: (a) calibration
period and (b) validation period.

### 3.1.2 Model selection for future groundwater level simulation and uncertainty analysis

To select the best performing model, the simulation results of the calibration and validation period were used to
calculate information criteria statistics. The posterior probability ($p_k$) was calculated using Eq. (8) for AIC,
AICc, BIC and KIC methods. The L2B4 model obtained the highest posterior probability of 1, whereas all other
models had negligible posterior probability for all information criteria as shown in Figure 6.

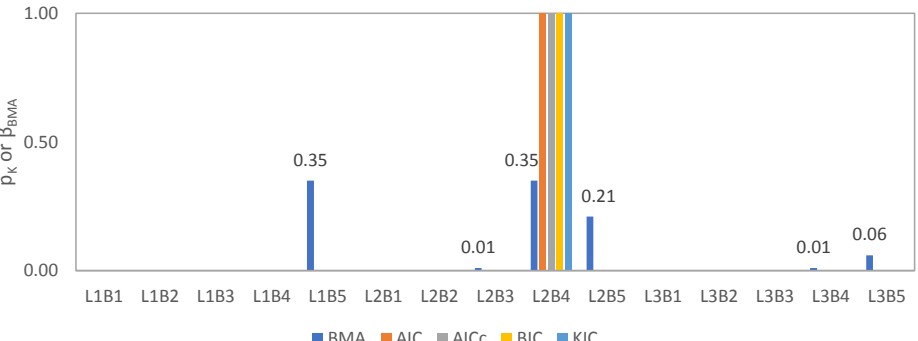


Figure 6: Posterior probability ($p_k$) and BMA maximum likelihood weight ($\beta_{BMA}$) of alternative models
calculated using 10 years of data. The value above the bar represents the maximum likelihood Bayesian weight.
One of the objectives was to estimate future groundwater levels using model averaging. Ten years (1990–1999)
of monthly simulated groundwater levels of the alternative models and observed data of 50 observation wells
were used as training data in MODELAVG to estimate the maximum likelihood BMA weight ($\beta_{BMA}$) of each



alternative model. The long training period was selected so that a reliable BMA weight can be estimated for
climate change impact analysis.
The performance evaluation statistics of BMA mean prediction along with the best model and median is shown
in supplementary materials (Table SM-6). The best model was selected based on the information criteria ranking.
The prediction of BMA method obtained better performance in all evaluation criteria than the best model and
ensemble median for both periods. The results are in line with the findings of Ye et al., (2004) and Poeter and
Anderson (2005).
During the training period, the 95% prediction interval covers about 85% of observed data, and the average
spread of the 95% prediction interval is 6.23 m. The maximum likelihood BMA weight ($\beta_{BMA}$) of all alternative
models is shown in Figure 6. It is observed that models L1B5 and L2B4 obtained higher $\beta_{BMA}$ than other models.
The model L2B4 has both maximum posterior model probability and higher $\beta_{BMA}$. It is noteworthy that the L1B5
model obtained significant $\beta_{BMA}$ as it had a comparatively poor performance in both calibration and validation
period compared to most of the other models. One possible cause could be the relatively better performance of
the one-layered model in the model boundary area.
Figure 6 shows that only three models (L1B5, L2B4, L2B5) together correspond to 91% of the total weight and
another three models (L2B3, L3B4, L3B5) correspond to 8% of the total weight. The rest of the models had no
significant contribution. The models having low $\beta_{BMA}$ can be excluded from the analysis to minimize the
calculation time and effort (Vrugt, 2016). Therefore, models L1B5, L2B4 and L2B5 were selected to predict
future groundwater levels under different scenarios. Ultimately, $\beta_{BMA}$ was recalculated using the prediction of
those selected models and the new $\beta_{BMA}$ of L1B5, L2B4 and L2B5 was 0.35, 0.39 and 0.26, respectively. During
this recalculation, the 95% prediction interval covers about 82% of observation data meaning exclusion of 12
models resulted in a loss of only 3% of observed data.
**3.2 Climate change impact on precipitation, temperature and evapotranspiration**
Figure 7a shows the changes in the monthly precipitation amount. Small positive changes in monthly
precipitation amounts are observed for the wet season. For the dry season, in contrary, the changes are less
consistent: decreasing precipitation amounts are found for April and December while March display a significant
increase. The effect of the greenhouse gas scenario (GHS) on the monthly precipitation amount changes is
shown by Figure 7b. One would expect increasing/decreasing change signals under increasing GHSs. This uni-
directional behavior is, however, limited to the months July, August, September and November. Most likely,





2035 is situated before the time of emergence, whereby the effect of the increasing GHS remains mainly masked
by noise inherent to the internal climate variability (Hawkins and Sutton, 2012). This, moreover, indicates that
the months July, August, September and November are most likely more sensitive to the GHSs compared to the
other months.

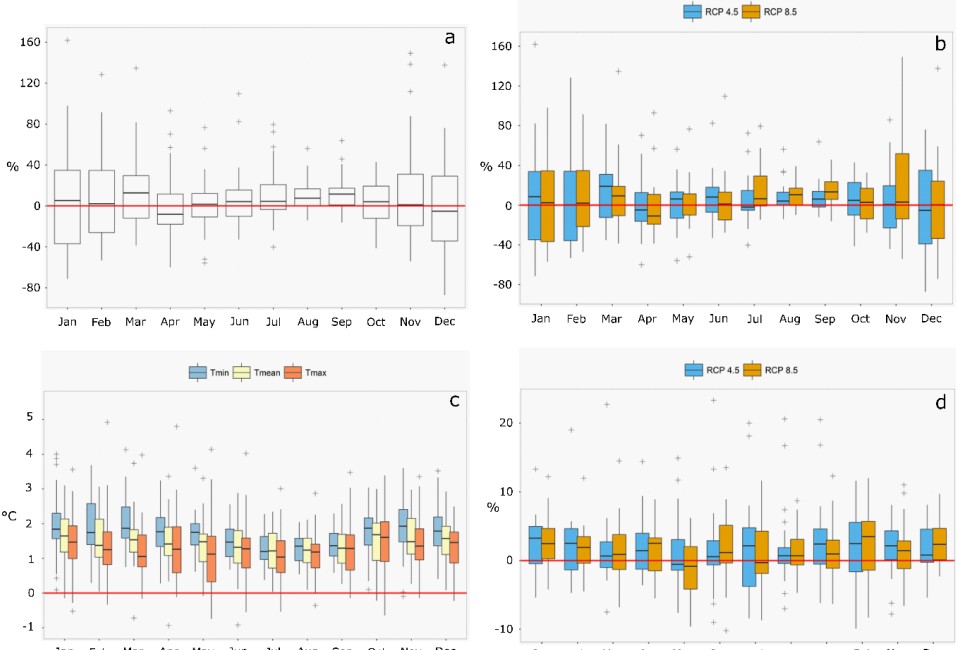


Figure 7: Climate impact signal for all selected climate models (1975 – 2035): (a) relative changes in monthly
precipitation amount (all GHS combined), (b) relative changes in monthly precipitation amount as function of
the GHSs, (c) absolute changes in monthly minimum, mean and maximum daily temperature (all GHSs
combined), and (d) relative changes in potential evapotranspiration as function of the GHSs.
Figure 7c presents the climate scenarios for minimum, mean and maximum daily temperature. Generally, higher
increases in minimum and mean daily temperatures are projected during the wet season. An inter-comparison
between the different variables shows, furthermore, higher changes for the minimum daily temperature than for
the mean and maximum daily temperature.
The changes in monthly potential evapotranspiration are shown in Figure 7d. Except for May, increases are
observed for all months. For some months, the changes seem not sensitive to the GHS. Changes for the months



March, April, June, October and December seem particularly sensitive to the GHS. Similar as for the
precipitation results, a possible explanation can be found in the "time of emergence" concept.
The climate change signals for a representative month in the dry and wet season are included in supplementary
materials (Table SM-8).
**3.3 Climate change impact on groundwater recharge**
The changes in the monthly groundwater recharge due to climate change are highly uncertain (Figure 8a). Like
precipitation, small increasing changes in monthly groundwater recharge are observed for the wet season. For the
dry season, in contrary, the changes are less consistent. The majority of the global climate model runs project
generally an increasing groundwater recharge. However, for April and December, significant decreases are
noted. The effect of the GHSs on the monthly groundwater recharge changes is shown by Figure 8b. The months
July, August, September and November seem to be more sensitive to the GHSs compared to the other months.
For both RCP 8.5 and RCP 4.5, April and December show decreasing changes in monthly groundwater recharge.

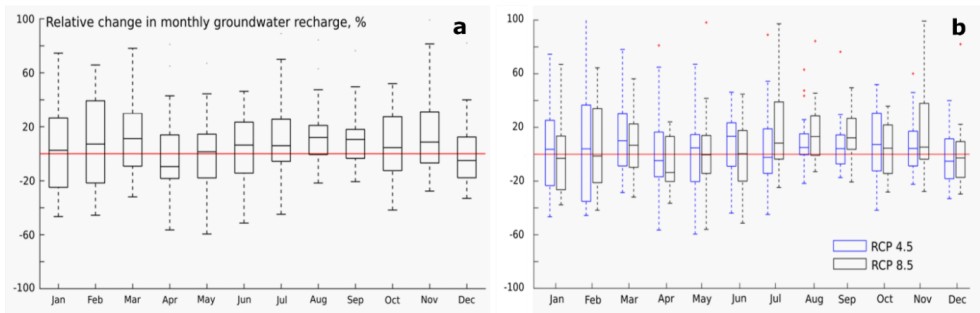


Figure 8: Change in groundwater recharge due to climate change: (a) relative changes in monthly groundwater
recharge (all GHS combined), (b) relative changes in monthly groundwater recharge as a function of the GHSs.
Projected spatial variation of the mean groundwater recharge change between the future and the baseline period
due to climate change is presented in Figure 9. Spatial variation is observed only for two extreme recharge
scenarios: high recharge scenario is indicating maximum recharge at each time step among all the ensembles and
low recharge scenario is indicating minimum recharge. Both for April and September, the high recharge scenario
shows a zero to positive change in groundwater recharge, while the low recharge scenario shows a zero to
negative change in groundwater recharge. No clear spatial trends are observed in the change of groundwater
recharge. In the high recharge scenario, mean monthly groundwater recharge would increase by 25 mm (April)
and 100 mm (September). In the low recharge scenario, mean monthly groundwater recharge would decrease by





16 mm (April) and 35 mm (September). Crosbie et al. (2010), also, reported that changes in groundwater
recharge due to climate change are uncertain.

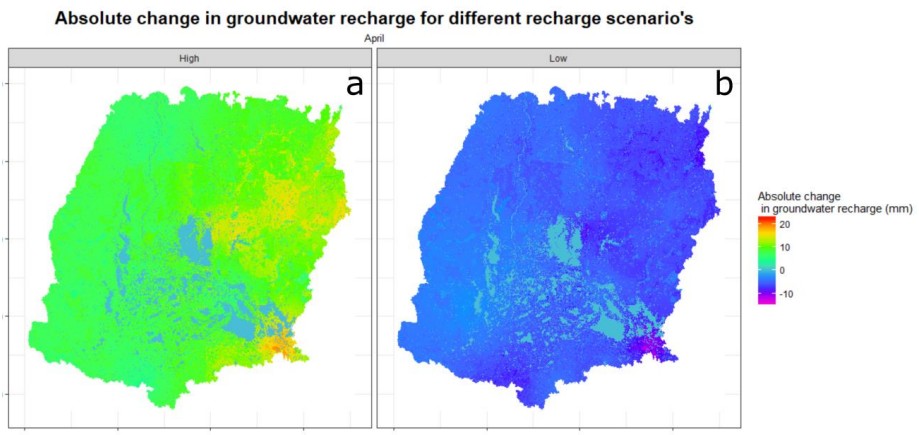

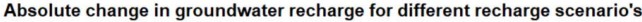

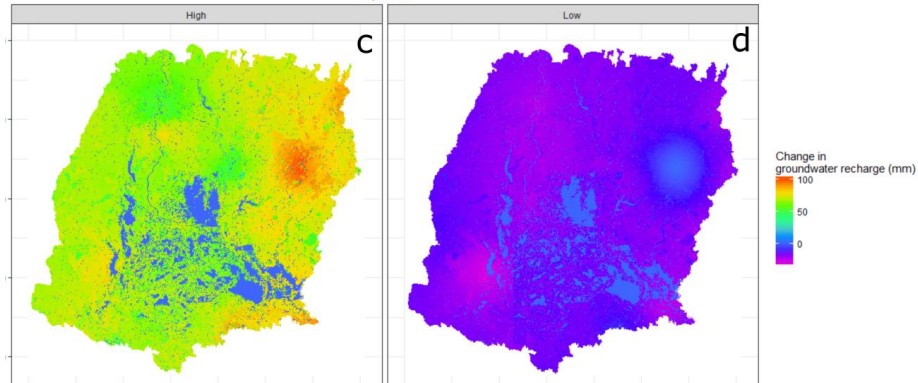


Figure 9: Spatial variation of mean groundwater recharge change due to climate change for (a) high recharge
scenario in April, (b) low recharge scenario in April, (c) high recharge scenario in September and (b) low
recharge scenario in September.
**3.4 Future groundwater level analysis**
The baseline and future groundwater levels were simulated using three selected groundwater flow models
(L1B5, L2B4, L2B5). Then, the model average was calculated by Eq. (10) using simulated groundwater levels
and the maximum likelihood Bayesian weight of the respective groundwater flow models. The change in
groundwater level for different scenarios is discussed below.



### 3.4.1 Baseline groundwater level simulation

Groundwater levels in the baseline scenario show a decreasing trend. The mean decreasing rate of groundwater level is 0.18 m/year (Sen's slope). The summary of the trend analysis for 50 observation wells is shown in supplementary materials (Table SM-9). The calculated decreasing rate varies spatially and ranges between 0.05 to 0.49 m/year. Mustafa et al. (2017a) studied observed groundwater level data of the same study area and reported that the average groundwater level dropped by 4.5–4.9 m over the last 29 years at a rate of 0.15–0.17 m/year. The annual groundwater level fluctuation of 3 to 5 m in the baseline scenario is also supported by the findings of Shamsudduha et al. (2009). Overall, the simulated groundwater levels correspond well with the findings of other researchers for the baseline period. Therefore, the simulated groundwater level of the baseline period was used for comparison with the simulated groundwater levels of the future scenarios.

### 3.4.2 Impact of climate change on groundwater level

Impact of climate change on groundwater level is highly uncertain in the study area (Figure 10a). The uncertainty ranges of the change in mean monthly groundwater level due to different GCMs and GHSs obtained from the three selected conceptual groundwater flow models are presented with the box-plot for each month. Climate change could increase the mean monthly groundwater level by up to 2.5 m and could decrease by 0.5 m. However, the SDGs suggest 0-0.5 m increase in groundwater level due to climate change. The impact of climate change seems higher from May to September than from October to April. This seasonal variation of climate change impact can be explained by the precipitation pattern of the study area (Supplementary materials: Figure SM-2a). Large precipitation amounts occur from May to October in Bangladesh, so that climate change has a higher impact on this period. Uncertainty of groundwater level due to climate change is highest from June to December. The precipitation pattern can also explain the monthly variation of climate change impact uncertainty. Groundwater levels increase more during the rainy season in a high recharge scenario (high precipitation), but in a low recharge scenario, groundwater levels decrease due to the lack of recharge in the rainy seasons. Therefore, the uncertainty band increases in this period for extreme scenarios. Similar to precipitation and groundwater recharge, the effect of the GHSs are not very significant on groundwater level changes (Figure 10b). Most of the GCMs project that the increase of groundwater level would be higher for RCP 8.5 compared to RCP 4.5 for all months.





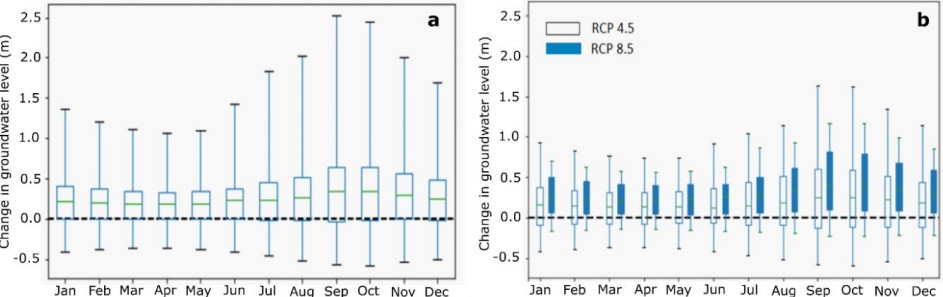


Figure 10: Mean monthly change of groundwater levels in the simulated future period (2026-2047) compared to
the baseline period (1980-2006) due to climate change: (a) all GHS combined, (b) as a function of the GHSs.
The impact of climate change on groundwater level also varies spatially. The projected impact of climate change
on groundwater level is relatively higher in the southwestern part (Figure 11) although this pattern does not
correspond to the spatial pattern of groundwater recharge (Figure 9). This can be explained by the effect of the
river on groundwater level. In a high recharge scenario mean monthly groundwater level would increase up to 4
m (April) and 8 m (September). However, in a low recharge scenario, mean monthly groundwater level would
decrease up to 1.6 m. Overall, the impact of climate change on groundwater level was not linear.





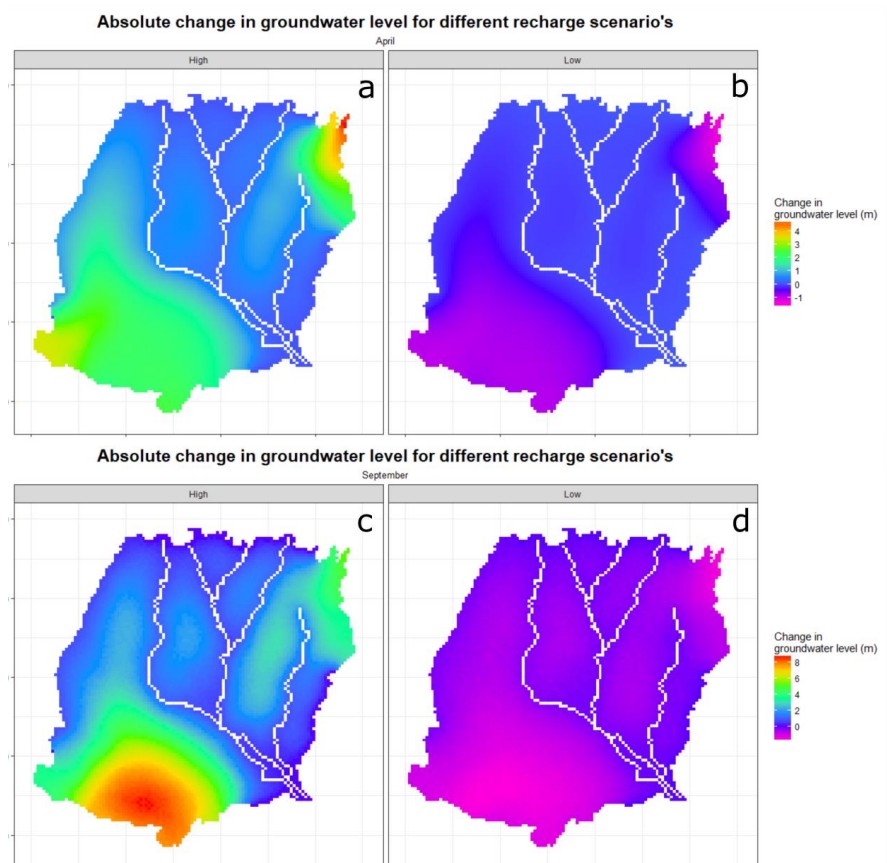


Figure 11: Spatial variation of mean groundwater level change due to climate change for the (a) high recharge scenario in April, (b) low recharge scenario in April, (c) high recharge scenario in September, and (b) low recharge scenario in September.

### 3.4.3 Future groundwater level under different abstraction scenarios

The mean monthly groundwater level for the $P_{Linear}$ abstraction scenario decreases about 10 to 14 m compared to the baseline period (Figure 12a). The scenario of $P_{Constant}$ resulted in a 4 to 7 m decrease in groundwater level (Figure 12b). For the 30% reduced ($P_{Reduced\_30}$) abstraction scenario, the mean groundwater level would decrease about 1.5 to 3.8 m (Figure 12c). Even for the 50% reduced ($P_{Reduced\_50}$) abstraction scenario, the mean groundwater level would decrease about 1.0 to 1.5 m (Figure 12d). Groundwater abstraction in the study area has to be reduced by 60% compared to the groundwater abstraction rate in 2010, to keep a sustainable groundwater level (Figure 12e). This indicates that the groundwater abstraction rate of 2010 is much higher than the future recharge potential. The situation will be worse if the current increasing groundwater abstraction trend continues.





A spatial variation in groundwater level change for different abstraction scenarios was also observed. In a low
recharge scenario, even for a 30 % reduced ($P_{Reduced\_30}$) abstraction scenario, groundwater level decreased about
14 m in the southwestern part of the study area. In a high recharge scenario, on the other hand, groundwater level
increased about 2 m in the northeastern part of the study area for the $P_{Reduced\_30}$ abstraction scenario. The results
also show that 50% lower groundwater abstraction than the 2010-rate is not enough to stop groundwater level
declining in the southwestern part of the study area.

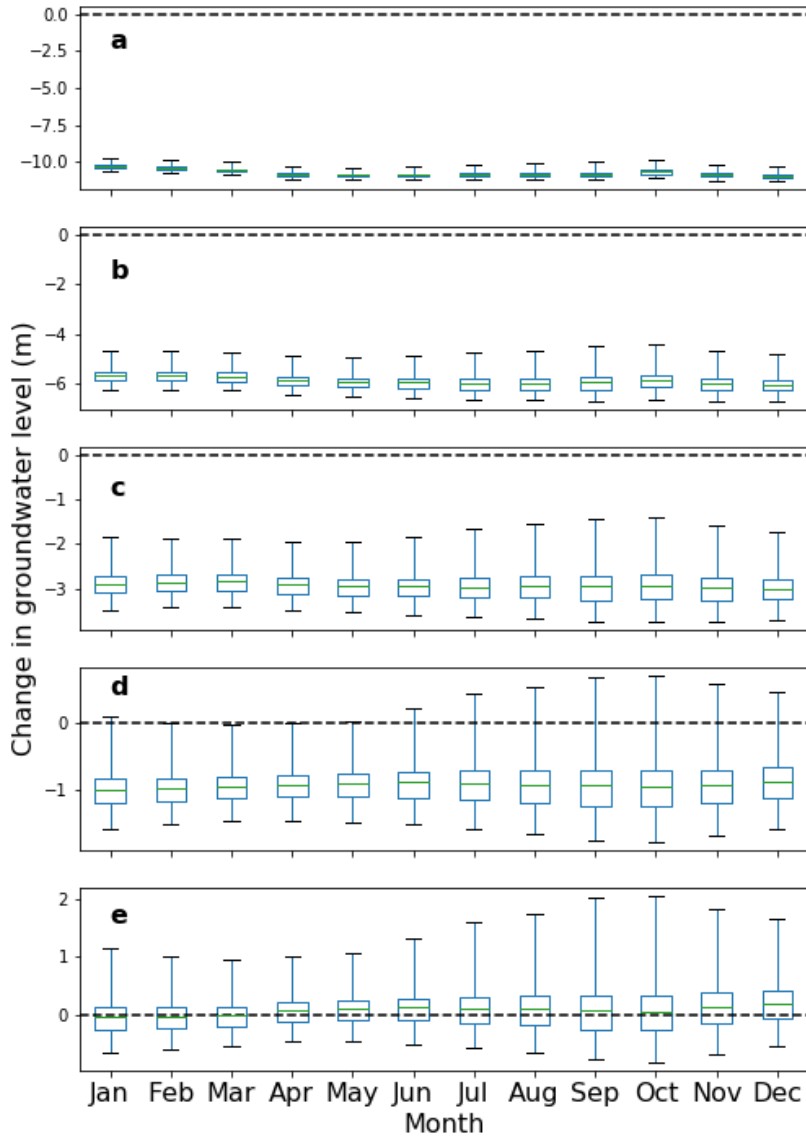





Figure 12: Monthly mean change in groundwater levels in the simulated future period (2026-2047) compared to
the baseline period (1985-2006) due to groundwater abstraction: (a) for $P_{Linear}$ abstraction scenario; (b) for
$P_{Constant}$ abstraction scenario; (c) for 30 % reduced ($P_{Reduced\_30}$) abstraction scenario; (d) for 50 % reduced
($P_{Reduced\_50}$) abstraction scenario and (e) for 60 % reduced ($P_{Reduced\_60}$) abstraction scenario.
The summary of annual groundwater level trend analysis of 50 observation wells for the high and low recharge
scenario and for different abstraction scenarios ($P_{Linear}$, $P_{Constant}$, and $P_{Reduced\_30}$) is shown in Table 2. Only the
significant ($p < 0.05$) trends were considered in this analysis. Scenario $P_{Constant}$ and $P_{Reduced\_30}$ have a mean
decreasing rate that is two to three times higher than the baseline scenario. Therefore, proper groundwater
abstraction policy is necessary to maintain sustainable use of this resource.
Table 2: The summary of annual groundwater level trend statistics of 50 observation wells for the baseline
(1985–2006) and simulated future (2026–2047) period under different abstraction scenarios ($P_{Linear}$, $P_{Constant}$,
$P_{Reduced\_30}$) and recharge scenarios (Low, High).

|  | Baseline period | Simulated future period | | | | | |
|  |  | $P_{Linear}$ | | $P_{Constant}$ | | $P_{Reduced\_30}$ | |
|  |  | Low | High | Low | High | Low | High |
| Statistics | Slope (m/year) | | | | | | |
| Mean | -0.18 | -1.10 | -1.02 | -0.50 | -0.47 | -0.37 | -0.30 |
| Maximum | -0.05 | -0.06 | -0.06 | -0.03 | -0.04 | -0.04 | -0.09 |
| Minimum | -0.49 | -3.89 | -3.71 | -1.88 | -1.54 | -1.13 | -0.79 |
| Median | -0.15 | -0.39 | -0.38 | -0.37 | -0.35 | -0.27 | -0.18 |
| Standard deviation | 0.11 | 1.23 | 1.12 | 0.51 | 0.40 | 0.29 | 0.25 |


**3.5 Sources of uncertainty in groundwater level prediction**
**3.5.1 Alternative conceptual model (CHMs) uncertainty**
The 95% prediction intervals of the three best performing models are shown in Figure 13a. The average spread
of the 95% prediction interval of the three alternative CHMs was about 3 m with a maximum spread of about 16
m. It is observed that the spread of the prediction interval is wider for low and high groundwater levels. This is
not surprising as the one-layered model overestimates low groundwater levels and underestimates high
groundwater levels in most of the observation wells. The wide uncertainty band of the alternative CHMs
indicates that the use of a single model in groundwater levels prediction may lead to biased conclusions.





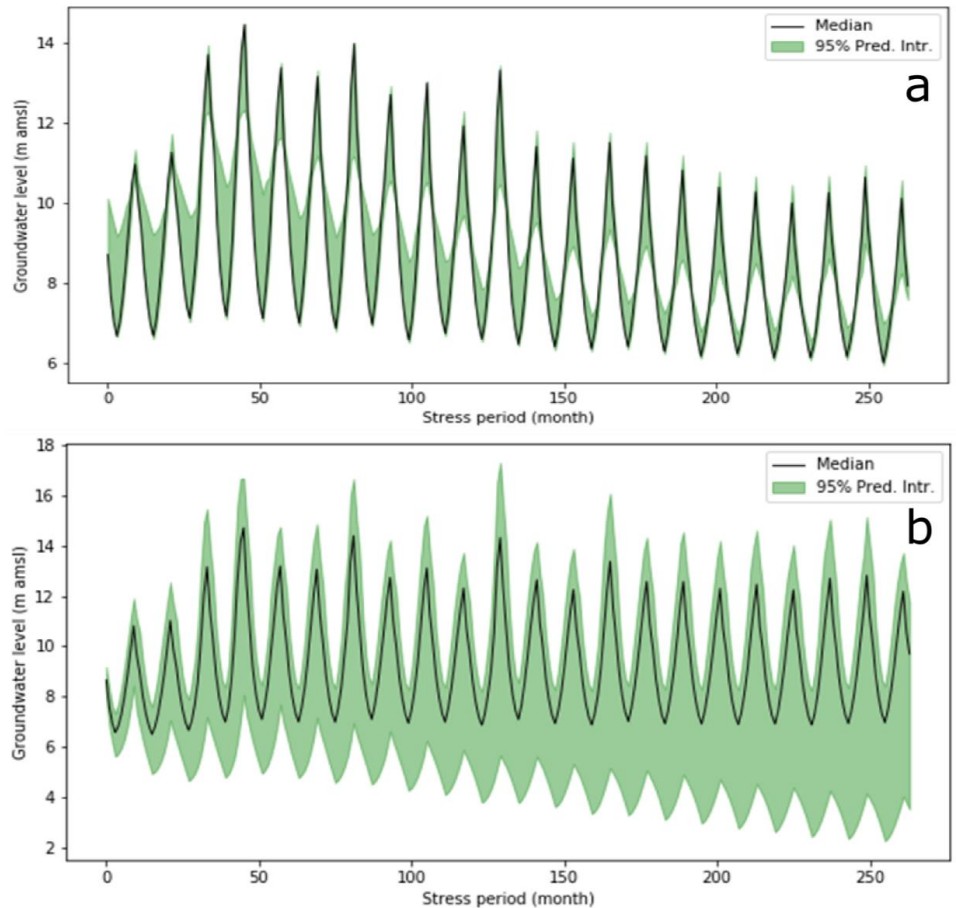


Figure 13: The 95% prediction interval of groundwater level of a representative observation well (BOG001) for
(a) different conceptual models and (b) different abstraction scenarios.

**3.5.2 Recharge scenarios uncertainty**

The average spread of the 95% prediction interval due to recharge scenarios is 1.11 m with a maximum of 6.07
m. The predictive uncertainty due to the recharge scenario is higher during periods with high groundwater levels
and recharge. Although the mean uncertainty resulting from recharge scenarios is relatively lower than for other
sources of uncertainty, there is large temporal and spatial variation in groundwater level prediction due to
recharge scenarios (as described in section 3.4.2). The recharge scenarios were developed using future climate
scenarios of different climate models so that the uncertainty from recharge scenarios represents the uncertainty
from climate scenarios in groundwater levels prediction. This uncertainty analysis suggests that all possible
climate scenarios should be considered to predict groundwater levels with a reliable uncertainty band.




### 3.5.3 Abstraction scenarios uncertainty


The 95% prediction interval of groundwater level for different abstraction scenarios increases with time (Figure
13b). The average spread of the 95% prediction interval is 8.38 m and the maximum is 43 m. The uncertainty of
groundwater level related to the abstraction scenario is very high.

### 3.5.4 Comparison of sources of uncertainties


The uncertainties due to alternative CHMs, recharge scenarios and abstraction scenarios are compared (Figure
14). The spread of the prediction interval of groundwater levels resulting from different CHMs, recharge
scenarios and abstraction scenarios was estimated using Eq. (13), (14) and (15), respectively. The contribution of
each source was calculated based on the median value of the spread of the prediction interval. The contribution
of an individual source is calculated as the ratio of the median value of the spread of the prediction interval for
the respective source to the median value of the spread of the prediction interval for the total uncertainty. The
abstraction scenarios are the dominant source of the total uncertainty in groundwater level prediction in this
overexploited aquifer. About 68% of the total uncertainty arises from the abstraction scenarios. CHM uncertainty
contributed about 23% of total uncertainty. This result is in agreement with the findings by Rojas et al. (2008).
They reported CHM uncertainty contributions up to 30%. In this case, the alternative CHM uncertainty
contribution is higher than the recharge scenario uncertainty contribution, including the greenhouse gas scenario,
climate model and stochastic climate uncertainty contributions. Goderniaux et al. (2015) reported that
uncertainty related to the calibration of hydrological models can be more important than uncertainty related to
climate models in groundwater modeling. The uncertainty due to recharge scenarios was relatively lower than
the other sources but the uncertainty arising from recharge scenarios was very high in the southwestern part of
the study area (described in section 3.4.2). Hence, use of a single model or single recharge or abstraction
scenario may lead to biased estimation of groundwater levels. Therefore, a multi-model and multi-scenario
approach should be used for reliable groundwater levels prediction.

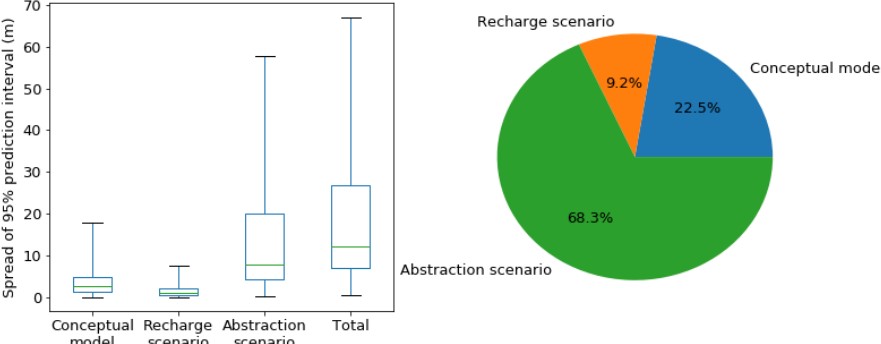


Figure 14: Comparison of uncertainties arising from alternative conceptual models, recharge scenarios and

abstraction scenarios. The recharge scenario uncertainty includes the greenhouse gas scenario uncertainty, the

climate model uncertainty and the stochastic uncertainty.

**4 Conclusions**

The main objective of this study was to quantify groundwater level prediction uncertainty in climate change

impact studies using an ensemble of representative concentration pathways, global climate models, multiple

alternative CHMs and abstraction. In this study, 15 alternative CHMs, 22 climate model runs for representative

concentration pathways 4.5 and 8.5 (in total 44 climate model runs) and 5 groundwater abstraction scenarios

were used to achieve this aim. The BMA technique was used to predict reliable groundwater level using

predictions of alternative CHMs.

It was observed that different conceptual groundwater models (CHMs) can simulate significantly different

groundwater levels due to differences in the number of layers and the boundary conditions. The simple one-

layered models were unable to simulate seasonal variation, but had a relatively better performance close to the

model boundaries than the other multi-layered models. The three-layered models were more detailed, but the

performance was not superior to the two-layered models. The performance of the two-layered models was

mostly better than the one-layered and three-layered models.

Ranking of models differed in the calibration and validation period. The best model in the calibration period only

got the 4th rank in the validation period suggesting the importance of the use of multiple CHMs for reliable

prediction.





The impact of groundwater abstraction on groundwater levels is very high. For 2026–2047, the groundwater
level would decline about 5 to 6 times faster than in the baseline period (1985–2006) if the current increasing
groundwater abstraction trend continues. Even with a 30% lower groundwater abstraction rate compared to the
2010-rate, the mean monthly groundwater level would decrease by up to 14m in the southwestern part of the
study area. Groundwater abstraction has to be reduced by 60% compared to the 2010-rate to keep groundwater
level sustainable. This indicates that the groundwater abstraction rate of 2010 was far higher than recharge
potential.
The differences in groundwater abstraction scenarios were the dominant source of uncertainty in groundwater
level prediction. The uncertainty due to alternative CHMs was also found to be significant and higher than the
uncertainty from the recharge scenarios. The uncertainty due to different recharge scenarios was very high in
southwestern part of study area. Therefore, use of a single model and/or single recharge and abstraction scenario
can lead to biased groundwater levels prediction.
This study suggests that a multi-model approach should be used in groundwater level prediction to avoid biased
estimation of groundwater levels. The BMA is probably the most suitable technique for developing a multi-
model average based on the best available data and future alternative scenarios. This study recommends that the
uncertainty due to alternative CHMs, recharge and abstraction scenarios should be considered in future
groundwater levels prediction.
**Acknowledgements**
We acknowledge the World Climate Research Programme's Working Group on Coupled Modelling, which is
responsible for CMIP, and we thank the climate modeling groups for producing and making available their
model output. The 5[th] author obtained a PhD scholarship from the Fund for Scientific Research (FWO)-Flanders.
This financial support is gratefully acknowledged.

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
