# Peer review of "Multi-model approach to quantify groundwater level prediction uncertainty using an ensemble of global climate models and multiple abstraction scenarios"

_Hydrology and Earth System Sciences, 2018_

## Referee Comment (RC1) · Anonymous Referee #1 · 20 Jan 2019

General comments:

The aim of the paper is to make a prediction of a future groundwater level, and to quantify the uncertainty of multiple sources of the models. This is achieved by using multiple conceptual hydrogeological models, climate scenarios and abstraction scenarios. I think the authors conducted a challenging project and present worthwhile results. A relatively simple hydrogeological model is applied which makes that the results have to be judged to that background. The paper has a clear structure and is well written.

0.88
0.93

[Figure]

Specific comments:

Line 273 The magnitude of the river bed conductance is given as 0.18 mˆ2/s (~15500 mˆ2/d). It is unclear what this quantity means. Usually, in MODFLOW the river bed conductance depends on the river length (L) and width (L) within a grid cell, and the (vertical) hydraulic conductivity (L/T) and the thickness (L) of the river bed. This yields a value with dimension (Lˆ2/T). This is also the dimension of the given conductance, instead of the expected dimension (L/T).

I ask the authors to explain the interpretation of this quantity.

Line 304 The model is calibrated using PEST. The values of the calibrated parameters are given in the supplementary materials in Table SM-2. The calibrated values of the L1 models are 6.00E-3 m/s (518 m/d) and 4.45E-3 m/s (384 m/d) which seem to be unrealistic high values for the described subsurface. The same order of magnitude holds for the second layer of the L2 models, and for the third layer of the L3 models.

Many calibrated parameters are set to the upper boundary of the parameter range. This suggests that the calibrated values could not reach the real optimum, or that conceptual problems in the models prevent a good calibration.

From these observations it may be concluded that the calibration of the hydrogeological model needs more attention. The achieved results, as described in the paper, have to be judged with in relation to the quality of the hydrogeological models.

I ask the authors to add a discussion of the quality of the calibration, and to explain the magnitude of the conductivity values and their validity in the model.

I suggest the authors to add in the discussion an improvement of the calibration in a future study.

Line 480 The RMSE and the variance are both used to test the goodness of fit of the models. In table SM-5 and SM-6, however, all RMSE values are exactly equal to the square root of the variance. The description of the variance in line 319 also seems to be

the same as the calculation of the RMSE. This suggests that there is no added value to use both measures to judge the quality of the models. Are the authors convinced about the correctness of the implementation of these measures? Or are the calculations of both measures inherently equal?

Please make clear what the value of the variance is or, in the case of equality of both measures, I would suggest to remove the presentation of one of the measures (RMSE or variance) from the results.

Another presented performance measure is the PBIAS in Eq. 2. This equation is applied to the observed and calculated groundwater levels. Since groundwater levels are measured against an arbitrary reference level I think the PBIAS is not a suitable measure to apply on these values. The numerator of the formula of PBIAS is not affected by the choice of the reference level but the denominator is. The PBIAS measure seems more suitable for quantities without an arbitrary reference level, like fluxes.

I ask the authors to make clear why PBIAS is a good performance indicator in the current study and why it can be used, or to replace it by another indicator or, if they agree with my objections, to remove it from the article.

Line 496 The authors describe the cause of the outliers in Fig. 5. It is not explicitly mentioned which observations the authors call the outliers, but it seems to be the observations beyond the 95% interval. Obviously, about 5% of the observations will lie beyond the 95% interval. The presented graph does not have extreme outliers, relatively to the total data cloud. More important is to what extent a difference between observed and calculated values is accepted in this study.

I ask the authors to make clear what they consider the acceptable difference between observed and calculated values, or which acceptable interval.

Line 562 In Fig. 7c the temperature changes calculated in the different scenarios are presented. Herein, the Tmax is lower (instead of higher) depicted than the Tmean and

[Figure]

Tmin, which is confusing.

Please explain what these values do represent?

Line 548 and Line 575 In these lines the period 'dry season' is mentioned. It would help the reader to repeat here which months are considered the dry season.

Technical corrections:

Line 65: first occurrence of CHMs should be singular

Line 74 increasing -> increasingly

Lines 86 abbreviation GHS is explained, Line 87 GHG is used

The words 'groundwater level' is often written as singular, where it should be plural.

I would suggest to add in long sentences commas (",") for readability.

---

## Referee Comment (RC2) · Anonymous Referee #2 · 25 Jan 2019

This paper deals with uncertainties in groundwater level predictions due to greenhouse gas scenarios, climate models, conceptual hydrogeological models (CHMs) and groundwater abstraction scenarios. To achieve this aim, ensemble of alternative CHMs, recharge and abstraction scenarios were used. The study confirms Bayesian Model Averaging (BMA) is the most suitable technique designed both to develop multi-model ensemble approach and to help account for the uncertainty inherent in the model selection process. The topic of the note lies within the aims and scope of Hydrology and Earth System Sciences and deals with a topic of considerable interest.

[Figure]

Multi-model approaches can be profitably associated with sensitivity analysis in order to answer the following questions: for a given set of measurements, which conceptual picture of the physical processes, as embodied in a mathematical model or models, is most appropriate? What are the most valuable space-time locations for measurements, depending on the model selected? How is model parameter uncertainty propagated to model output, and how does this propagation affect model calibration? Recent examples of methods to combine sensitivity-based calibration and model selection have been presented in literature right in the context of groundwater modelling. I suggest to the authors to deepen this topic since, at this stage, the paper does not introduce significant scientific advances respect to the state of art. It's true that typically parametric uncertainty dominates in literature with respect to the uncertainty related to models and scenarios. Nevertheless, this is not enough to make the paper self contained. This is a general evaluation on the study that brought me to the decision that the work still needs major revisions to make it acceptable for publication.

Specific suggestions to improve the quality of the paper are listed below.

1. I suggest to add a schematic representation of the system investigated for the sake of clarity. This will help identifying the calibration parameters in one/two/three-layered models respectively. 2. With the goal of facilitating the understanding of the study, it may be worthwhile to insert the equations used in the analysis and not just references. 3. Please reword paragraphs 2.7 "Future groundwater recharge scenario" providing more details about model adopted and 2.10 "Data analysis" explaining more clearly the procedure followed. 4. Improve the quality/size of the figures to highlight the results of the analysis

Minor points:

5. Check line 65, "CHMs", remove "s". 6. Check line 192, in "step" a "s" is missing; 7. Check line 424, reference is missing; 8. Check Line 480, reference is missing.

[Figure]

580, 2018.

---

## Author Comment (AC2) · 9 Mar 2019

The comment was uploaded in the form of a supplement:
https://www.hydrol-earth-syst-sci-discuss.net/hess-2018-580/hess-2018-580-AC2-supplement.pdf

---

## Author Response (AR1)

Response to Editor and Reviewer comments

**HESS-2018-580**

**Title: Multi-model approach to quantify groundwater level prediction uncertainty using an ensemble of global climate models and multiple abstraction scenarios**
Authors: Syed M. Touhidul Mustafa*, M. Moudud Hasan, Ajoy Kumar Saha, Rahena Parvin Rannu, Els Van Uytven, Patrick Willems and Marijke Huysmans

We would like to thank the editor and reviewers for reviewing our manuscript very carefully and for their constructive comments. We have considered all the comments and changed the manuscript accordingly. Below is a list of our responses to the editor and reviewer comments (comments in italic, answers in regular font).

Please kindly note that the line numbers in the responses refer to the numbering in the revised manuscript, unless specified.

**Editor:**

*General comments:*
*You did only partly answered to the reviewers comments. Therefore, provide a revised version of the manuscript with your suggested revisions and improve your answers on the following comments:*

RESPONSE: AGREE AND CHANGES MADE

We have clarified our answers to some of the comments as requested and adapted the manuscript and response letter accordingly.

**1. - comment 2, rev.1: you have to explain why the calibrated parameters reach the boundary. It makes the calibration questionable.**

RESPONSE: AGREE AND CHANGES MADE (lines: 336 - 352).

The following additional explanation about the calibration has been added in the manuscript (lines: 336 - 352).

The optimized value of specific yield varies between 0.17 and 0.35 for different conceptual models. The results are in line with previous finding of specific yield values in the area which indicate that specific yield in the study area varies between 0.08 and 0.32, having higher values in the southern part of the Barind area (Jahan et al., 1994; Mustafa et al., 2018). However, the optimized value of specific yield for some conceptual models are equal to the upper boundary of the pre-defined parameter range. This could be because of the simplified representation of hydrogeological layers and properties of the system defined in some of the conceptual models. However, even with different conceptual models, the optimized value of specific yield is equal to the upper boundary of the parameter range, indicating that the calibrated values of the specific yield could not reach the real optimum. This could be because of uncertain groundwater abstraction and recharge data in this study area. Mustafa et al. (2018) has proven that groundwater abstraction and groundwater recharge data in space and time in this study area are highly uncertain. They have also reported that input uncertainty (uncertainties arising from groundwater abstraction and recharge) has a significant impact on the specific yield values. However, in this study, uncertainty of the input data has not been considered. Additionally, spatial and seasonal variability of the groundwater abstraction has not been considered in this study. This might be another reason for the high specific yield value. Further improvement of model calibration would require additional and more reliable groundwater abstraction and groundwater recharge data, such as time series of pumping discharge from individual wells and exact locations of all abstraction wells.

Jahan, C.S., Mazumder, Q.H., Ghose, S.K., Asaduzzaman, M., 1994. Specific yield evaluation: Barind area, Bangaladesh. Geol. Soc. India 44(3), 283–290.

Mustafa, S.M.T., Nossent, J., Ghysels, G., Huysmans, M., 2018. Estimation and impact assessment of input and parameter uncertainty in predicting groundwater flow with a fully distributed model. Water Resour. Res. 54(9), 6585-6608.

*2. - comment 3, rev.1: you have to explain why the variances are all equal.*

RESPONSE: AGREE AND CHANGES MADE

By definition, RMSE values are equal to the square root of the variance. Therefore, there is no added value in using both measures to judge the quality of the models and the variance value and its calculation procedure have been removed from the manuscript.

*3. - comment 5, rev. 1: Improve the discussion of the calibration results. Some of the differences are important. Are they located in some specific locations?*

RESPONSE: AGREE AND CHANGES MADE (lines: 527 – 541)

We believe that the observations wells with high differences between modelled and observed hydraulic heads are located close to pumping wells about which the information about their locations and pumping discharge is highly uncertain in this study area. In order to discuss this issue in the manuscript, the relevant section in the original manuscript has been updated with additional information and explanation as follows.

Figure 5 shows the scatter plot for model L2B5. One of the possible causes of the observed differences is the spatial and temporal variation in groundwater abstraction. The zone-wise spatially distributed groundwater abstraction rate was one of the most important input data in this study. In reality, groundwater abstraction varies spatially within those zones. Agricultural and industrial areas abstract more groundwater than wetlands or forest areas. Moreover, groundwater abstraction rate also varies in time following cropping seasons and precipitation patterns. However, an average constant groundwater abstraction rate was assumed for six months (from November to April) in the model. The difference between observed and simulated are high for some observation wells. Those observation wells might be located near to abstraction wells. For observation wells close to groundwater abstraction wells, drawdown by groundwater abstraction could affect observed groundwater heads. This spatial and temporal difference in actual groundwater abstraction and modeled groundwater abstraction causes spatial and temporal variation in simulated and observed groundwater levels. The simplified representation of hydrogeological layers and properties could be also a possible cause of the differences between simulated and observed groundwater levels. For simplification, the aquifer was assumed homogeneous but in reality the aquifer is heterogeneous and this may affect groundwater flow in the aquifer. Also, measurement errors in observation data may influence model performance.

**Reviewer #1:**

*General comments:*

*The aim of the paper is to make a prediction of a future groundwater level, and to quantify the uncertainty of multiple sources of the models. This is achieved by using multiple conceptual hydrogeological models, climate scenarios and abstraction scenarios. I think the authors conducted a challenging project and present worthwhile results. A relatively simple hydrogeological model is applied which makes that the results have to be judged to that background. The paper has a clear structure and is well written.*

 **Specific comments:**

*1. Line 273 The magnitude of the river bed conductance is given as 0.18 mˆ2/s (~15500 mˆ2/d). It is unclear what this quantity means. Usually, in MODFLOW the river bed conductance depends on the river length (L) and width (L) within a grid cell, and the (vertical) hydraulic conductivity (L/T) and the*

*thickness (L) of the river bed. This yields a value with dimension (L^2/T). This is also the dimension of the given conductance, instead of the expected dimension (L/T).*

*I ask the authors to explain the interpretation of this quantity.*

RESPONSE: AGREE AND CHANGES MADE (lines: 273 - 280)

Riverbed conductance is indeed defined as a lumped parameter in MODFLOW defined as:

CRIV $= \frac{K_{riv} \times L \times W}{M_{riv}}$

Where, CRIV= Riverbed hydraulic conductance ($L^2T^{-1}$)

$K_{riv}$ = riverbed sediment hydraulic conductivity ($LT^{-1}$)

L = Length of the river within a grid cell (L)

W = Width of the river within a grid cell (L)

$M_{riv}$ = Thickness of the riverbed within a grid cell (L).

From the equation, it is clear that riverbed hydraulic conductance depends on grid-size, riverbed sediment hydraulic conductivity and thickness of the riverbed. Mehl and Hill (2010) have reported that riverbed conductance depends heavily on grid-size of the model. Hence, direct interpretation on the quantity of riverbed hydraulic conductance is not straightforward.

This additional explanation and motivation were added to the manuscript (lines: 273 - 280).

Mehl, S., & Hill, M. C. (2010). Grid-size dependence of Cauchy boundary conditions used to simulate stream–aquifer interactions. Advances in water resources, 33(4), 430-442.

*2. Line 304 The model is calibrated using PEST. The values of the calibrated parameters are given in the supplementary materials in Table SM-2. The calibrated values of the L1 models are 6.00E-3 m/s (518 m/d) and 4.45E-3 m/s (384 m/d) which seem to be unrealistic high values for the described subsurface. The same order of magnitude holds for the second layer of the L2 models, and for the third layer of the L3 models.*

*Many calibrated parameters are set to the upper boundary of the parameter range. This suggests that the calibrated values could not reach the real optimum, or that conceptual problems in the models prevent a good calibration.*

*From these observations it may be concluded that the calibration of the hydrogeological model needs more attention. The achieved results, as described in the paper, have to be judged with in relation to the quality of the hydrogeological models.*

*I ask the authors to add a discussion of the quality of the calibration, and to explain the magnitude of the conductivity values and their validity in the model.*

*I suggest the authors to add in the discussion an improvement of the calibration in a future study*

RESPONSE: AGREE AND CHANGES MADE (lines: 317 – 331; 336 – 352 and 786 – 790)

The optimized horizontal hydraulic conductivity of the one-layered models varies between $4.45 \times 10^{-03}$ m/s and $6.00 \times 10^{-03}$ m/s. This high value of horizontal hydraulic conductivity corresponds to well-sorted coarse sand and gravel (Fetter, 2001). We consider these values to be realistic since a major portion of the aquifer consists of coarse sand and coarse sand with gravel. The average horizontal hydraulic conductivity of Bengal basin found by Michael & Voss (2009b) was also high ($5 \times 10^{-04}$ m/s). They also reported that based on the drill-log analysis horizontal hydraulic conductivity of Bengal basin may varies from $6 \times 10^{-06}$ m/s to $3.00 \times 10^{-03}$ m/s. The area of the Bengal basin is about $2.8 \times 10^{5}$ km$^2$, but the study area is only a small part of the Bengal basin. Therefore, it is possible that the horizontal hydraulic conductivity is relatively higher in our study area. Bonsor et al. (2017) have also reported in their review report that aquifer materials in the Bengal basin are highly permeable. Mustafa et al. (2018) have also reported that average horizontal hydraulic conductivity of this study area is high and around $2.5 \times 10^{-3}$ and $4.5 \times 10^{-3}$ m/s.

Additionally, spatial variability of horizontal hydraulic conductivity has not been considered in this study. We consider an average horizontal conductivity for all individual layers. This might be another reason for high horizontal hydraulic conductivity.

We agree with the reviewer's view that the calibration of the hydrogeological model needs more attention to constrain model parameters. Model calibration using a global optimization method is more reliable than a optimization tool like PEST. However, Mustafa et al. (2018) have also reported in their research paper on uncertainty estimation and impact assessment using global optimization that average horizontal hydraulic conductivity of this study area is high and around $2.5 \times 10^{-3}$ - $4.5 \times 10^{-3}$ m/s.

All the details on the magnitude of hydraulic conductivity and model calibration processes have been added to the revised manuscript (lines: 317 – 331).

The following additional explanation about the calibration has been added in the manuscript (lines: 336 - 352).

The optimized value of specific yield varies between 0.17 and 0.35 for different conceptual models. The results are in line with previous finding of specific yield values in the area which indicate that specific yield in the study area varies between 0.08 and 0.32, having higher values in the southern part of the Barind area (Jahan et al., 1994; Mustafa et al., 2018). However, the optimized value of specific yield for some conceptual models are equal to the upper boundary of the pre-defined parameter range. This could be because of the simplified representation of hydrogeological layers and properties of the system defined in some of the conceptual models. However, even with different conceptual models, the optimized value of specific yield is equal to the upper boundary of the parameter range, indicating that the calibrated values of the specific yield could not reach the real optimum. This could be because of uncertain groundwater abstraction and recharge data in this study area. Mustafa et al. (2018) has proven that groundwater abstraction and groundwater recharge data in space and time in this study area are highly uncertain. They have also reported that input uncertainty (uncertainties arising from groundwater abstraction and recharge) has a significant impact on the specific yield values. However, in this study, uncertainty of the input data has not been considered. Additionally, spatial and seasonal variability of the groundwater abstraction has not been considered in this study. This might be another reason for the high specific yield value. Further improvement of model calibration would require additional and more reliable groundwater abstraction and groundwater recharge data, such as time series of pumping discharge from individual wells and exact locations of all abstraction wells.

We have also added to the discussion section that in this study, alternative conceptual models have been calibrated using PEST. However, different calibration methods can result in different calibrated model parameters. Hence, further studies could be conducted using different calibration methods (e.g. global parameters optimization methods). We also advice that more field data would be collected, such as reliable groundwater abstraction data, river flow information, spatially distributed horizontal hydraulic conductivity and detailed information about the boundary conditions. (lines: 786 – 790).

Bonsor, H.C., MacDonald, A.M., Ahmed, K.M., Burgess, W.G., Basharat, M., Calow, R.C., et al., 2017. Hydrogeological typologies of the Indo-Gangetic basin alluvial aquifer, South AsiaTypologies. Hydrogeol. J. 1–30.

Fetter, C.W., 2001. Applied Hydrogeology. 4th Edition, Prentice Hall, Upper Saddle River, 2, 8.

Jahan, C.S., Mazumder, Q.H., Ghose, S.K., Asaduzzaman, M., 1994. Specific yield evaluation: Barind area, Bangaladesh. Geol. Soc. India 44(3), 283–290.

Michael, H.A., Voss, C.I., 2009b. Controls on groundwater flow in the Bengal Basin of India and Bangladesh: regional modeling analysis. Hydrogeol. J. 17, 1561.

Mustafa, S.M.T., Nossent, J., Ghysels, G., Huysmans, M., 2018. Estimation and impact assessment of input and parameter uncertainty in predicting groundwater flow with a fully distributed model. Water Resour. Res. 54(9), 6585-6608.

*3. Line 480 The RMSE and the variance are both used to test the goodness of fit of the models. In table SM-5 and SM-6, however, all RMSE values are exactly equal to the square root of the variance. The description of the variance in line 319 also seems to be the same as the calculation of the RMSE. This suggests that there is no added value to use both measures to judge the quality of the models. Are the authors convinced about the correctness of the implementation of these measures? Or are the calculations of both measures inherently equal?*

*Please make clear what the value of the variance is or, in the case of equality of both measures, I would suggest to remove the presentation of one of the measures (RMSE or variance) from the results.*

RESPONSE: AGREE AND CHANGES MADE

By definition, RMSE values are equal to the square root of the variance. Therefore, there is no added value in using both measures to judge the quality of the models and the variance value and its calculation procedure have been removed from the manuscript.

*4. Another presented performance measure is the PBIAS in Eq. 2. This equation is applied to the observed and calculated groundwater levels. Since groundwater levels are measured against an arbitrary reference level I think the PBIAS is not a suitable measure to apply on these values. The numerator of the formula of PBIAS is not affected by the choice of the reference level but the denominator is. The PBIAS measure seems more suitable for quantities without an arbitrary reference level, like fluxes.*

*I ask the authors to make clear why PBIAS is a good performance indicator in the current study and why it can be used, or to replace it by another indicator or, if they agree with my objections, to remove it from the article.*

RESPONSE: AGREE AND CHANGES MADE

PBIAS and its calculation procedure have been removed from the manuscript.

*5. Line 496 The authors describe the cause of the outliers in Fig. 5. It is not explicitly mentioned which observations the authors call the outliers, but it seems to be the observations beyond the 95% interval. Obviously, about 5% of the observations will lie beyond the 95% interval. The presented graph does not have extreme outliers, relatively to the total data cloud. More important is to what extent a difference between observed and calculated values is accepted in this study.*

*I ask the authors to make clear what they consider the acceptable difference between observed and calculated values, or which acceptable interval.*

RESPONSE: AGREE AND CHANGES MADE (lines: 527 – 541)

There are indeed no extreme outliers. To avoid confusion, the relevant sentences have been updated by removing the word "outliers".

We believe that the observations wells with high differences between modelled and observed hydraulic heads are located close to pumping wells about which the information about their locations and pumping discharge is highly uncertain in this study area. In order to discuss this issue in the manuscript, the relevant section in the original manuscript has been updated with additional information and explanation as follows.

Figure 5 shows the scatter plot for model L2B5. One of the possible causes of the observed differences is the spatial and temporal variation in groundwater abstraction. The zone-wise spatially distributed groundwater abstraction rate was one of the most important input data in this study. In reality, groundwater abstraction varies spatially within those zones. Agricultural and industrial areas abstract more groundwater than wetlands or forest areas. Moreover, groundwater abstraction rate also varies in time following cropping seasons and precipitation patterns. However, an average constant groundwater abstraction rate was assumed for six months (from November to April) in the model. The difference between observed and simulated are high for some observation wells. Those observation wells might be located near to abstraction wells. For observation wells close to groundwater abstraction wells, drawdown by groundwater abstraction could affect observed groundwater heads. This spatial and temporal difference in actual groundwater abstraction and modeled groundwater abstraction causes spatial and temporal variation in simulated and observed groundwater levels. The simplified representation of hydrogeological layers and properties could be also a possible cause of the differences between simulated and observed groundwater levels. For simplification, the aquifer was assumed homogeneous but in reality the aquifer is heterogeneous and this may affect groundwater flow in the aquifer. Also, measurement errors in observation data may influence model performance.

*6. Line 562 In Fig. 7c the temperature changes calculated in the different scenarios are presented. Herein, the Tmax is lower (instead of higher) depicted than the Tmean and Tmin, which is confusing.*

*Please explain what these values do represent?*

RESPONSE: AGREE AND CHANGES MADE (lines: 579 – 580; 595 – 597)

Figure 7 shows the changes in monthly climatic parameters between the control and scenario period ranging between 1961-1990 and 2021-2050, respectively. Figure 7c shows the absolute changes in monthly minimum, mean and maximum daily temperature between the control and scenario period.

Here, the figures show that the changes in Tmax are lower compared to the changes of Tmean and Tmin.

This section has been updated with this additional clarification to avoid confusion.

*7. Line 548 and Line 575 In these lines the period 'dry season' is mentioned. It would help the reader to repeat here which months are considered the dry season.*

RESPONSE: AGREE AND CHANGES MADE (lines: 582 and 609)

Months considered for the dry season have been added.

**Technical corrections:**

*8. Line 65: first occurrence of CHMs should be singular*

RESPONSE: AGREE AND CHANGES MADE (line: 65)

The additional "S" has been removed from "CHMs".

*9. Line 74 increasing -> increasingly*

RESPONSE: AGREE AND CHANGES MADE (line: 74)

The word increasing has been replaced by increasingly.

*10. Lines 86 abbreviation GHS is explained, Line 87 GHG is used*

RESPONSE: AGREE AND CHANGES MADE (line: 87)

Line 87 of the original manuscript has been updated with GHS instead of GHG.

*11. The words 'groundwater level' is often written as singular, where it should be plural.*

RESPONSE: AGREE AND CHANGES MADE

This has been corrected.

*12. I would suggest to add in long sentences commas (",") for readability.*

RESPONSE: AGREE AND CHANGES MADE

Commas (",") have been added to the long sentences.

**Reviewer # 2**

***General comments:***

*This paper deals with uncertainties in groundwater level predictions due to greenhouse gas scenarios, climate models, conceptual hydrogeological models (CHMs) and groundwater abstraction scenarios. To achieve this aim, ensemble of alternative CHMs, recharge and abstraction scenarios were used. The study confirms Bayesian Model Averaging (BMA) is the most suitable technique designed both to develop multi-model ensemble approach and to help account for the uncertainty inherent in the model selection process. The topic of the note lies within the aims and scope of Hydrology and Earth System Sciences and deals with a topic of considerable interest.*

*Multi-model approaches can be profitably associated with sensitivity analysis in order to answer the following questions: for a given set of measurements, which conceptual picture of the physical processes, as embodied in a mathematical model or models, is most appropriate? What are the most*

*valuable space-time locations for measurements, depending on the model selected? How is model parameter uncertainty propagated to model output, and how does this propagation affect model calibration? Recent examples of methods to combine sensitivity-based calibration and model selection have been presented in literature right in the context of groundwater modelling. I suggest to the authors to deepen this topic since, at this stage, the paper does not introduce significant scientific advances respect to the state of art. It's true that typically parametric uncertainty dominates in literature with respect to the uncertainty related to models and scenarios. Nevertheless, this is not enough to make the paper self contained. This is a general evaluation on the study that brought me to the decision that the work still needs major revisions to make it acceptable for publication.*

RESPONSE: AGREE AND CHANGES MADE: ADDITIONAL EXPLANATION IN THE TEXT (lines: 791 – 796)

We agree with the reviewer that multi-model approaches are associated with sensitivity analysis in order to answer the following questions: for a given set of measurements, which conceptual picture of the physical processes, as embodied in a mathematical model or models, is most appropriate? What are the most valuable space-time locations for measurements, depending on the model selected? How is model parameter uncertainty propagated to model output, and how does this propagation affect model calibration?

However, the main objective of this study was not to identify the optimum parameters set or the best conceptual model structure. Our main objective was to evaluate the combined effect of conceptual hydro(geo)logical models (CHMs) structure, climate change and groundwater abstraction scenarios on future groundwater level prediction uncertainty. That is why we have incorporated all possible alternatives based on the available field data. Additionally, a separate study on the effect of input and parameter uncertainty has been published in Water Resources Research (Mustafa et al., 2018).

In order to highlight the important of sensitivity analysis, the following sentences have been added to the revised manuscript: "Keeping in mind that the complexity of hydrogeological models is increasing, further studies should be conducted on global sensitivity analysis (SA) to (i) identify the influential and non-influential parameters on the model prediction and (ii) better understand the importance of the different components of the complex model structure. Identification of influential parameters will play an important role in model parameterization and in reducing uncertainty due to overparameterization. The identification of non-influential parameters using SA will be a very important step in simplifying model structure".

Mustafa, S. M. T., Nossent, J., Ghysels, G., & Huysmans, M. (2018). Estimation and impact assessment of input and parameter uncertainty in predicting groundwater flow with a fully distributed model. Water Resources Research, 54(9), 6585-6608.

**Specific suggestions to improve the quality of the paper are listed below:**

*1) I suggest to add a schematic representation of the system investigated for the sake of clarity. This will help identifying the calibration parameters in one/two/three-layered models respectively.*

RESPONSE: AGREE AND CHANGES MADE (lines: supplementary materials: Table SM-1)

A schematic representation of the system with all details of the calibration parameters used in the one/two/three-layered models, including the number of parameters, has been added in the supplementary materials (Table SM-1).

*2) With the goal of facilitating the understanding of the study, it may be worthwhile to insert the equations used in the analysis and not just references.*

RESPONSE: AGREE AND CHANGES MADE: ADDITIONAL EXPLANATION ADDED TO THE TEXT

All the relevant equations are given in the manuscript (Equations: 1-15).

*3) Please reword paragraphs 2.7 "Future groundwater recharge scenario" providing more details about model adopted and 2.10 "Data analysis" explaining more clearly the procedure followed.*

RESPONSE: AGREE AND CHANGES MADE (lines: 421 – 422 and 490)

All the details about the model adopted for this study are explained in section 2.6 of the original manuscript. However, for the sake of clarity, the following sentence has been added in the section 2.7. "Details about the considered climate model runs for this study are explained in section 2.6 and they are listed in the supplementary materials (Table SM-7)".

Section 2.10 has been updated by adding the following sentence: "Details about the procedure followed for data analysis is explained in sections 2.4 to 2.9".

*4) Improve the quality/size of the figures to highlight the results of the analysis*

RESPONSE: AGREE AND CHANGES MADE

The quality of the figures (Figure 1, 2, 4, 5, 7, 8, 10, and 13) has been improved.

**Minor points:**

*5) Check line 65, "CHMs", remove "s".*

RESPONSE: AGREE AND CHANGES MADE (line: 65)

The additional "S" has been removed from "CHMs".

*6) Check line 192, in "step" a "s" is missing.*

RESPONSE: AGREE AND CHANGES MADE (line: 193)

The sentence has been updated by adding an additional "s".

*7) Check line 424, reference is missing.*

RESPONSE: AGREE AND CHANGES MADE (line: 460)

 A reference has been added.

*8) Check Line 480, reference is missing.*

RESPONSE: AGREE AND CHANGES MADE (line: 515)

A reference has been added.

[revised manuscript text omitted]